# Anaerobic thiosulfate oxidation by the *Roseobacter* group is prevalent in marine biofilms

Wei Ding[1,2], Shougang Wang[1], Peng Qin[3], Shen Fan[3], Xiaoyan Su[4], Peiyan Cai[2], Jie Lu[3], Han Cui[3], Meng Wang[4], Yi Shu[1], Yongming Wang[1], Hui-Hui Fu[4], Yu-Zhong Zhang [4,5], Yong-Xin Li [2,6] ✉ & Weipeng Zhang [3] ✉

Thiosulfate oxidation by microbes has a major impact on global sulfur cycling. Here, we provide evidence that bacteria within various *Roseobacter* lineages are important for thiosulfate oxidation in marine biofilms. We isolate and sequence the genomes of 54 biofilm-associated *Roseobacter* strains, finding conserved *sox* gene clusters for thiosulfate oxidation and plasmids, pointing to a niche-specific lifestyle. Analysis of global ocean metagenomic data suggests that *Roseobacter* strains are abundant in biofilms and mats on various substrates, including stones, artificial surfaces, plant roots, and hydrothermal vent chimneys. Metatranscriptomic analysis indicates that the majority of active *sox* genes in biofilms belong to *Roseobacter* strains. Furthermore, we show that *Roseobacter* strains can grow and oxidize thiosulfate to sulfate under both aerobic and anaerobic conditions. Transcriptomic and membrane proteomic analyses of biofilms formed by a representative strain indicate that thiosulfate induces *sox* gene expression and alterations in cell membrane protein composition, and promotes biofilm formation and anaerobic respiration. We propose that bacteria of the *Roseobacter* group are major thiosulfate-oxidizers in marine biofilms, where anaerobic thiosulfate metabolism is preferred.

The cycling of inorganic sulfur compounds by microbes is important biogeochemical progress. Sulfur transformation involves thiosulfate as an intermediate, which is abundant in marine environments[1,2]. It is well known that sulfur-oxidizing bacteria include diverse autotrophic bacteria from various groups, such as SUP05, SAR324, BS-GSO2, and Chlorobi[3]. These bacteria employ the *sox* multi-enzyme pathway for thiosulfate oxidation[4,5], and can grow autotrophically using thiosulfate as an electron donor for energy production[6]. The end-products of thiosulfate oxidation differ between different bacteria, as certain obligate organotrophic bacteria (e.g., *Pseudomonas stutzeri* and *Catenocossus thiosyclus*) oxidize thiosulfate incompletely to tetrathionate,

while other heterotrophs (e.g., *Bosea thiooxidans* and *Limnobacter thiooxidans*) oxidize thiosulfate to sulfate, the most oxidized sulfur form, in a similar pathway common among autotrophic sulfur-oxidizing bacteria[7]. First found in the lithoautotrophic bacteria *Paracoccus pantotrophus*[5,8], the sulfur oxidizing enzyme (SOX) system, a typical periplasmic multi-enzyme system, is the most well-known mediators of thiosulfate oxidation. Regarding electron acceptor utilization and niche distribution, aerobic thiosulfate oxidizers have been isolated from the deep chlorophyll maximum[9], coastal seawater[10], and coastal marine sediments[11], whereas anaerobic thiosulfate oxidizers have been recently reported in anoxic marine basins[12], oceanic oxygen

[1]College of Marine Life Sciences and MOE Key Laboratory of Marine Genetics and Breeding, Ocean University of China, Qingdao, China. [2]Department of Chemistry and The Swire Institute of Marine Science, The University of Hong Kong, Hong Kong, China. [3]Institute of Evolution & Marine Biodiversity, Ocean University of China, Qingdao, China. [4]Frontiers Science Center for Deep Ocean Multispheres and Earth System, Ocean University of China, Qingdao, China. [5]State Key Laboratory of Microbial Technology, Shandong University, Qingdao, China. [6]Southern Marine Science and Engineering Guangdong Laboratory, Guangzhou, China. ✉e-mail: yxpli@hku.hk; zhangweipeng@ouc.edu.cn

minimum zones[13], and hydrothermal vents[14]. In addition, environmental conditions affect the major products of thiosulfate production. For example, in coastal marine sediments, thiosulfate is oxidized to varying proportions of tetrathionate and sulfate, as well as elemental sulfur, depending on the sulfidic and oxygenic conditions[11].

As an important niche for many microbes, biofilms constitute 40% of ocean microbial biomass[15]. Many microbes form biofilms when they start to attach to any surface immersed in water, such as artificial materials, rock surfaces, animal skins, and marine snow[15,16]. A marine biofilm community can be composed of hundreds of microbial species, and structurally the biofilm is characterized by heterogeneity in terms of stratified oxygen concentration, nutrients, and signaling molecules[17]. Our recent studies[18,19] suggested that marine biofilms constitute a bank of unknown taxa and functions, containing over 7000 bacterial species not otherwise found in seawater microbiota. Moreover, marine biofilms are reported to be involved in important biogeochemical processes, such as converting dissolved organic matter, remineralizing particles, and passing nutrients from surface waters to deep oceans[20]. However, probably due to complex physical community structures and taxonomic compositions, thiosulfate oxidation in marine biofilms remains largely unknown, particularly in relation to the contributions of different taxa and their relevant metabolic characteristics. Specific questions are which bacteria are the greatest contributors to thiosulfate oxidation in marine biofilms and how they conduct this process.

Bacteria of the *Roseobacter* group (or the Roseobacteraceae family[21]), sharing > 89% identity in the 16 S rRNA gene, are heterotrophs found worldwide in marine ecosystems. The so far discovered *Roseobacter* group comprises 327 species and 128 genera[21], represented by *Ruegeria* (e.g, *Ruegeria pomeroyi* DSS-3[22], a model strain), *Phaeobacter* (well-known strains for the production of tropodithietic acid[23]), *Sulfitobacter* (widely-distributed strains in the global ocean[24]), and *Loktanella* (adapted to the polar regions[25]). Certain *Roseobacter* strains are reported to be associated with biofilms on surfaces of marine algae, invertebrates, and particles, although most of the isolated *Roseobacter* strains are free-living. For instance, analyses of the *Tara* ocean data suggested that *Ruegeria mobilis* strains occur in 40 and 6% of samples of the particle-associated fraction and the free-living fraction, respectively[26], implying their preference for a biofilm-associated lifestyle. Physiologically, *Roseobacter* strains are often known for their metabolic versatility, mainly involving carbon monoxide oxidation, aromatic compound degradation, secondary metabolic production, and oxidation of sulfur compounds[27–29]. The genomes of several isolated *Roseobacter* strains contain *sox* genes, although their ability to oxidize thiosulfate has not been experimentally demonstrated. Moreover, as thiosulfate oxidation is affected by the availability of electron acceptors, it is possible that the oxygen gradient in biofilms may promote the development of novel thiosulfate-oxidizing bacteria.

Here we hypothesized that *Roseobacter* strains represent important oxidizers of thiosulfate in marine biofilms. To test this hypothesis, we systematically studied the role of *Roseobacter* strains in thiosulfate oxidation in global ocean biofilms, beginning with large-scale bacterial isolation and screening. The metabolic features of 54 novel *Roseobacter* strains isolated from marine biofilms were predicted by complete genome sequencing. The in situ distribution and activity were explored by metagenomics and metatranscriptomics. Furthermore, physiological and biochemical experiments, transcriptomics, and proteomics demonstrated significant niche specificity in bacterial oxidation of thiosulfate.

## Results

### Novel *Roseobacter* strains isolated from marine biofilms

We isolated and cultured bacterial strains from biofilms on natural rocks immersed in coastal waters. After preliminary analyses of 16S

rRNA gene sequences generated by Sanger sequencing, more than 500 non-redundant strains were identified, including 54 distant strains affiliated with *Roseobacter* (hereafter referred to as biofilm *Roseobacter* strains). Then, the complete genomes of the 54 strains were sequenced using the PacBio sequencing technique. The basic information based on genomic analyses, including accession numbers, genome sizes, GC content, number of plasmids, number of open reading frames (ORFs), and number of rRNA and tRNA genes, is provided in Supplementary Data 1. Taxonomic classification based on whole-genome searching against the GTDB-TK database[30] divided the 54 strains into nine genera, including *Alliroseovarius*, *Jannaschia*, *Leisingera*, *Phaeobacter*, *Roseicyclus*, *Ruegeria*, *Sulfitobacter*, *Yoonia*, and one potentially new genus represented by the strain M382 (Supplementary Data 1). These strains were further classified into 16 species, including *Alliroseovarius crassostreae*, *Leisingera caerulea*, *Leisingera aquaemixtae*, *Phaeobacter inhibens*, *Sulfitobacter mediterraneus*, *Sulfitobacter pontiacus*, and 10 new species (Supplementary Data 1).

To study the evolutionary relationships between biofilm *Roseobacter* strains and previously reported *Roseobacter* strains, 95 complete genomes were downloaded from NCBI and used as references to build a phylogenetic tree with the 54 genomes documented in this study, using 31 conserved single-copy genes. These reference genomes included representative and well-studied members of the *Roseobacter* group such as *Planktomarina temperate* RCA23 and *R. pomeroyi* DSS-3. The tree pattern was found to be consistent with the taxonomic classification findings and indicated that our isolation and genome sequencing effort had identified several new lineages (Fig. 1). For example, M382 formed an independent branch while M385 and S2214 were situated relatively distant from the other strains. Notably, there were no reported complete genomes of *Leisingera* before the current study (Fig. 1).

As the GTDB-TK classification and the phylogenetic analysis are only based on conserved genes, the average nucleotide identity (ANI) based on whole-genome sequences was calculated. Pairwise comparison of the 54 genomes revealed ANI values ranging from 76.52% to 98.99% (Supplementary Fig. 1), indicating that they represent distinct strains. Comparison between M382 and its closely related reference strains *R. pomeroyi* DSS-3, *Ruegeria* sp. AD91A, and *Ruegeria* sp. THAF33 revealed lower identity values (78.6, 78.1, and 78.2, respectively) than the identities between the three referenced *Ruegeria* species (78.9, 79.4, and 91.0, respectively) (Supplementary Fig. 2), indicating that it probably represents a new genus. Together, we have discovered novel *Roseobacter* members and have expanded the genomic information on this bacterial group.

### Diverse plasmids and conserved *sox* gene clusters in the genomes of biofilm *Roseobacter* strains

With the complete genome sequences in hand, we next analyzed the genomic potentials of these biofilm-associated *Roseobacter* strains. We assumed that biofilm plasmids exemplified by the presence of a rhamnose operon[31] could be detected. Indeed, most of these bacteria harbored genes related to biofilm formation in their plasmids, although no rhamnose operon was detected (Supplementary Fig. 3). Genes responsible for colanic acid biosynthesis were detected in the plasmids of 10 *Leisingera* strains (Supplementary Fig. 3). The filamentous hemagglutinin gene was identified in the plasmids of three *Aliiroseovarius* strains and one *Leisingera* strain (Supplementary Fig. 3). Succinoglycan biosynthesis genes were identified in one *Leisingera* strain and two *Sulfitobacter* strains (Supplementary Fig. 3). The exopolysaccharide production gene *exoQ* was observed in a *Leisingera* strain (Supplementary Fig. 3). All these genes have been reported to contribute to biofilm formation in a variety of species[32–34]. To provide a detailed view of the biofilm plasmids, the map of one *Leisingera* sp. M527 plasmid was drawn as an example to show the presence of diverse pathways for polysaccharide biosynthesis and transport in a plasmid (Supplementary Fig. 4).

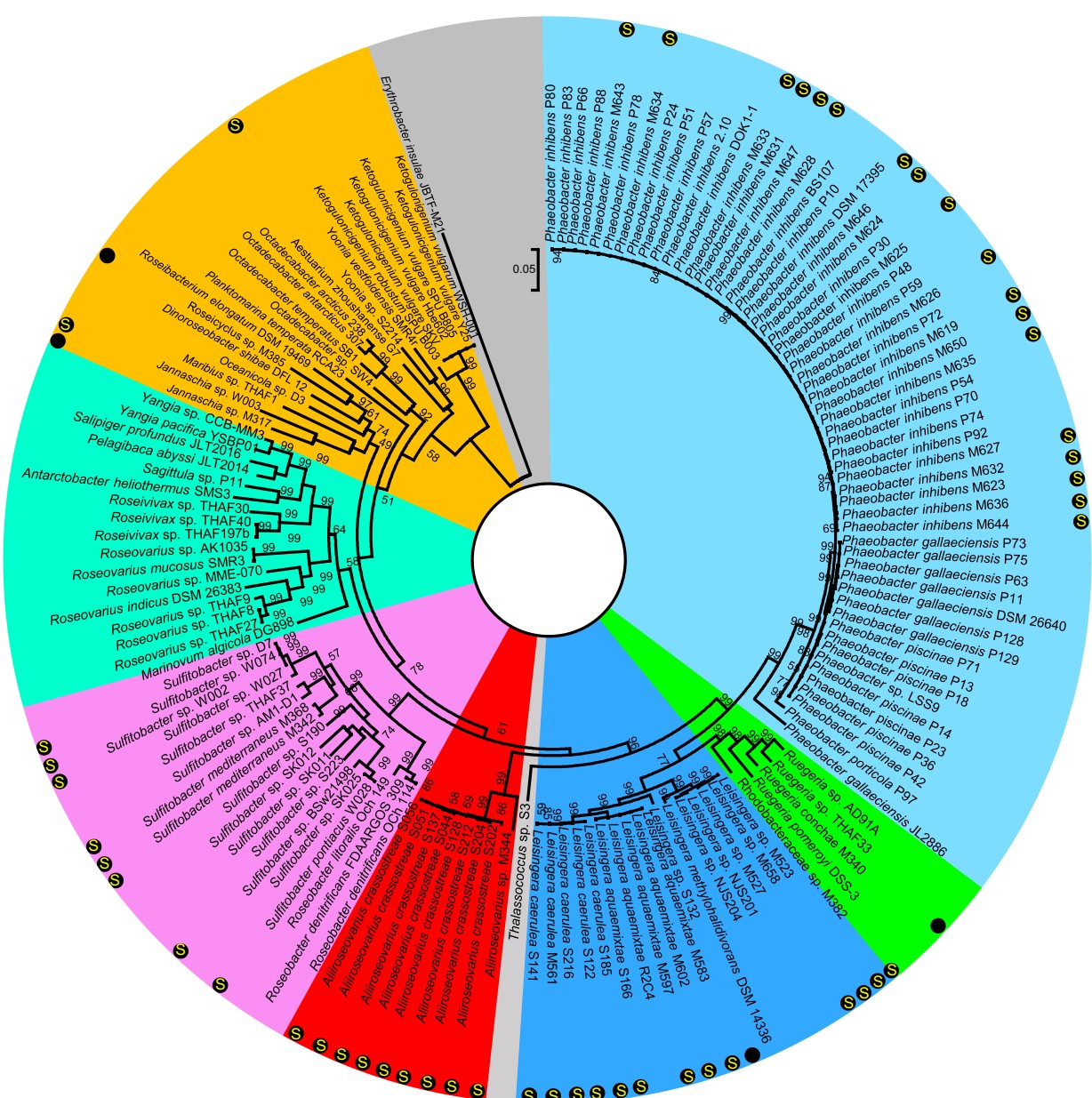

**Fig. 1 | Phylogenetic tree of the 54 biofilm-derived strains and 95 reference *Roseobacter* strains with complete genomes in NCBI.** Single-copy genes (*n* = 31) predicted from these genomes were concatenated to build the tree. The maximum-likelihood method based on the Jones−Taylor−Thornton matrix-based model was used. Bootstrap values derived from 500 replicated calculations are shown on tree branches. The 54 strains isolated from current study are labeled by black dots in the outline. "S" in the black dots indicates the presence of the *sox* gene cluster.

With regard to central carbon metabolism, components of the tricarboxylic acid (TCA) cycle were completely represented in all the genomes. Key enzymes of the Entner−Doudoroff (ED) pathway and the pentose phosphate (PP) pathway, such as glucose-6-phosphate 1-dehydrogenase (*zwf*, K00036) and glucose-6-phosphate isomerase (*pg*i, K01810), were identified in all the genomes. In contrast, the Embden−Meyerhof−Parnas (EMP) pathway was incomplete in the genomes of all *Phaeobacter*, *Ruegeria*, and *Sulfitobacter* strains, as well as M382, M385, and S2214, with the absence of the key enzymes 6-phosphofructokinase 1 (K00850) and 2 (K16370) (Supplementary Fig. 5). No genome possessed pathways for carbon dioxide fixation, suggesting a heterotrophic lifestyle. In terms of nitrogen metabolism, a variety of genes involved in denitrification, including *napA* (K02567), *napB* (K02568), *narG* (K00370), *narH* (K00371), *nirK* (K00368), or *nirS*

(K15864) were identified in 53 of the 54 genomes (Supplementary Fig. 6). The presence of these genes suggested a potential for respiration using nitrate or nitrite as electron acceptors. We also analyzed the potential availability of oxygen as an electron acceptor in respiration. All the genomes possessed diverse cytochrome metabolic genes, including those that use cytochrome cbb3 and those that use cytochrome c as substrates (Supplementary Fig. 7). It is known that cytochrome cbb3 oxidases, which have high oxygen affinity, are often used by aerobic microbes, whereas cytochrome c oxidases, which have low oxygen affinity, are found in anaerobic microbes[35]. Thus, the presence of both cytochrome cbb3 oxidases and cytochrome c oxidase implied the potential for both aerobic and anaerobic respiration.

The genome contents were further analyzed in terms of sulfur metabolism. Consistent with our hypothesis proposed in the

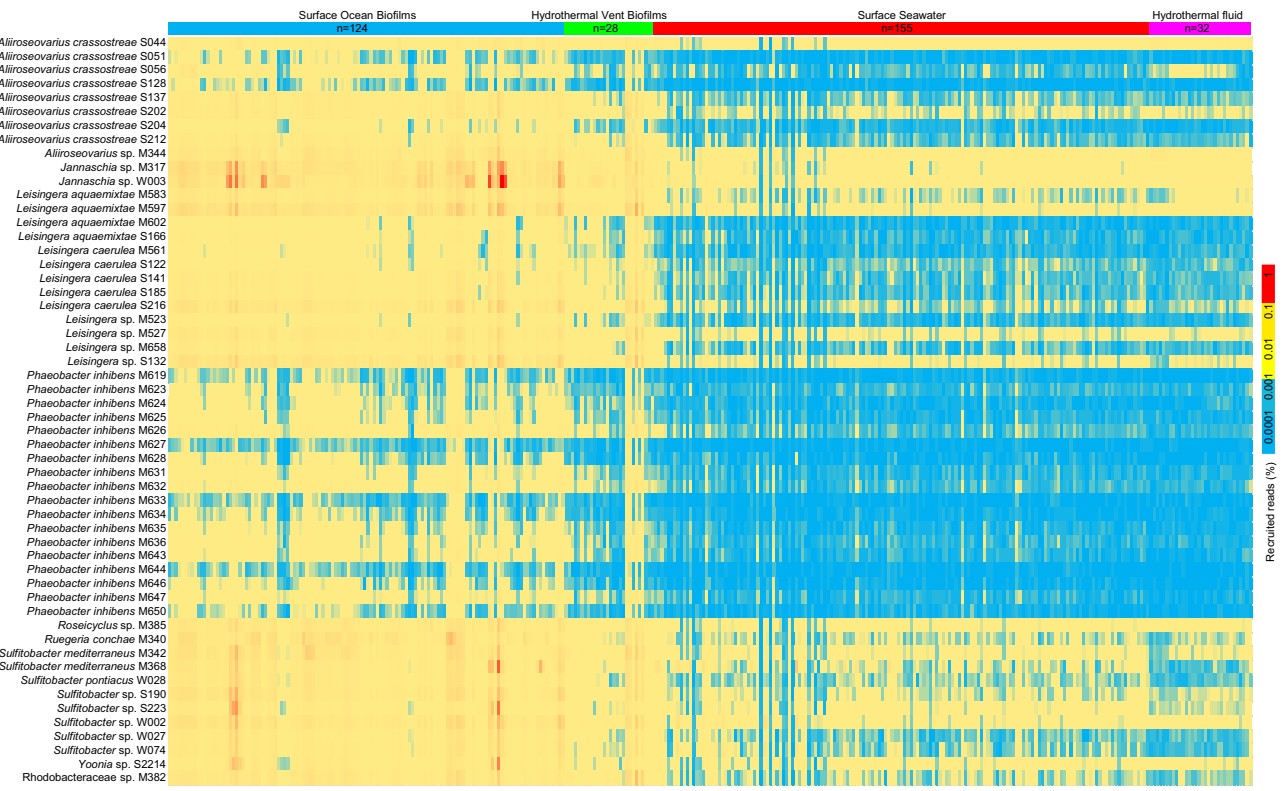

**Fig. 2 | Global distribution of *Roseobacter* strains in biofilm-associated and free-living microbiota.** The distribution pattern was drawn by mapping reads from 339 (152 biofilm and mat metagenomes versus 187 seawater and hydrothermal vent-fluid metagenomes) to chromosomes of the 54 biofilm-derived *Roseobacter* strains using BBMap (minimum alignment identity = 0.80). All the metagenomes were normalized to 1,000,000 reads with a read length of 101 bp. Source data are provided as a Source data file.

Introduction, diverse sulfur reduction or oxidation pathways were identified. The presence of *sat* (K00958) and *cys* (K00860, K00390, K00380, and K00381) genes suggested the capacity of assimilatory sulfate reduction, while the presence of sulfide: quinone oxidoreductases (K17218) suggested the potential of sulfide oxidation. In terms of thiosulfate oxidation, 50 of the 54 genomes possessed complete *sox* gene clusters, comprising *soxA* (K17222), *soxX* (K17223), *soxB* (K17224), *soxC* (K17225), *soxD* (K22622), *soxY* (K17226), and *soxZ* (K17227) (Supplementary Fig. 8). Sequential patterns of the *sox* gene cluster were drawn, revealing nearly identical patterns in all the strains except *Sulfitobacter* sp. S223, *Yoonia* sp. S2214, and Rhodobacteraceae sp. M382 (Supplementary Fig. 9). Moreover, the evolutionary path of the *sox* gene cluster was analyzed by comparing the functional gene tree based on concatenated *sox* genes (n = 7) with the species tree based on concatenated single-copy genes (n = 31), revealing similar tree topology patterns (Supplementary Fig. 10).

Together, the genomic analyses revealed that (1) the presence of novel biofilm plasmids in marine *Roseobacter* strains; (2) these *Roseobacter* strains may utilize various oxidants for respiration; (3) conservation of the *sox* gene cluster implies conserved functions associated with thiosulfate oxidation.

## Global distribution and niche specificity of biofilm *Roseobacter* strains as revealed by metagenomics

The prevalence of *sox* genes suggested that the biofilm *Roseobacter* strains are important thiosulfate oxidizers in global ocean biofilms, which motivated us to analyze their abundance in biofilm metagenomes collected across different ocean provinces and depths. To this end, 152 biofilm-associated microbiota (biofilms and mats) were obtained, including six newly collected biofilms in the present study (biofilm samples collected in Sep 2020, Nov 2020, Jan 2021, Mar 2021, May 2021, and Jul 2021, which were labeled as biofilm_1, biofilm_2, biofilm_3, biofilm_4, biofilm_5, and biofilm_6, respectively), 101 biofilms collected in eight locations of the surface ocean in our previous study[18], four microbial mats from surface oceans[36], eight plastic-surface biofilms[37], two metal-surface biofilms[38], three biofilms on seagrass root surfaces[39], two biofilms from the Lost City hydrothermal vent fields[40], four biofilms from the Old City hydrothermal vent fields[41], three mats from the Kallisti Limnes hydrothermal seafloor pool[42], and 19 hydrothermal vent chimney surface biofilms from three ocean provinces (Supplementary Data 2). In parallel, 187 metagenomes of free-living microbiota were analyzed and compared, including 155 surface seawater samples from the *Tara* Ocean project[43], and 32 hydrothermal vent fluid samples (Supplementary Data 2). The results showed extensive enrichment of *Roseobacter* strains in biofilm-associated microbiota, regardless of location, depth, substrate type, and time point for biofilm development (Fig. 2). The most abundant strain was *Jannaschia* sp. W003, which accounted for 1.88% of a biofilm formed on a plastic surface in the East China Sea (Fig. 2). The most enriched strains were *L. aquaemixtae* M602, *L. aquaemixtae* S166, and *Leisingera* sp. M523, which showed clearly contrasting distribution patterns between biofilm-associated and free-living samples (Fig. 2). *Phaeobacter* strains were only present in several biofilms and nearly undetectable in all the free-living microbiota (Fig. 2). Summarizing the total percentages of the 54 strains in different communities suggested that they accounted for a total of 0.12–6.32% (on average 1.00%) in the surface ocean biofilms, compared to only 0.03–0.38% (on average 0.08%) in the surface seawater samples. In parallel, they accounted for 0.07–3.38% (on average 0.48%) in the hydrothermal vent biofilms, compared to only 0.05–0.08% (on average 0.07%) in the hydrothermal vent fluid samples (Supplementary Fig. 11).

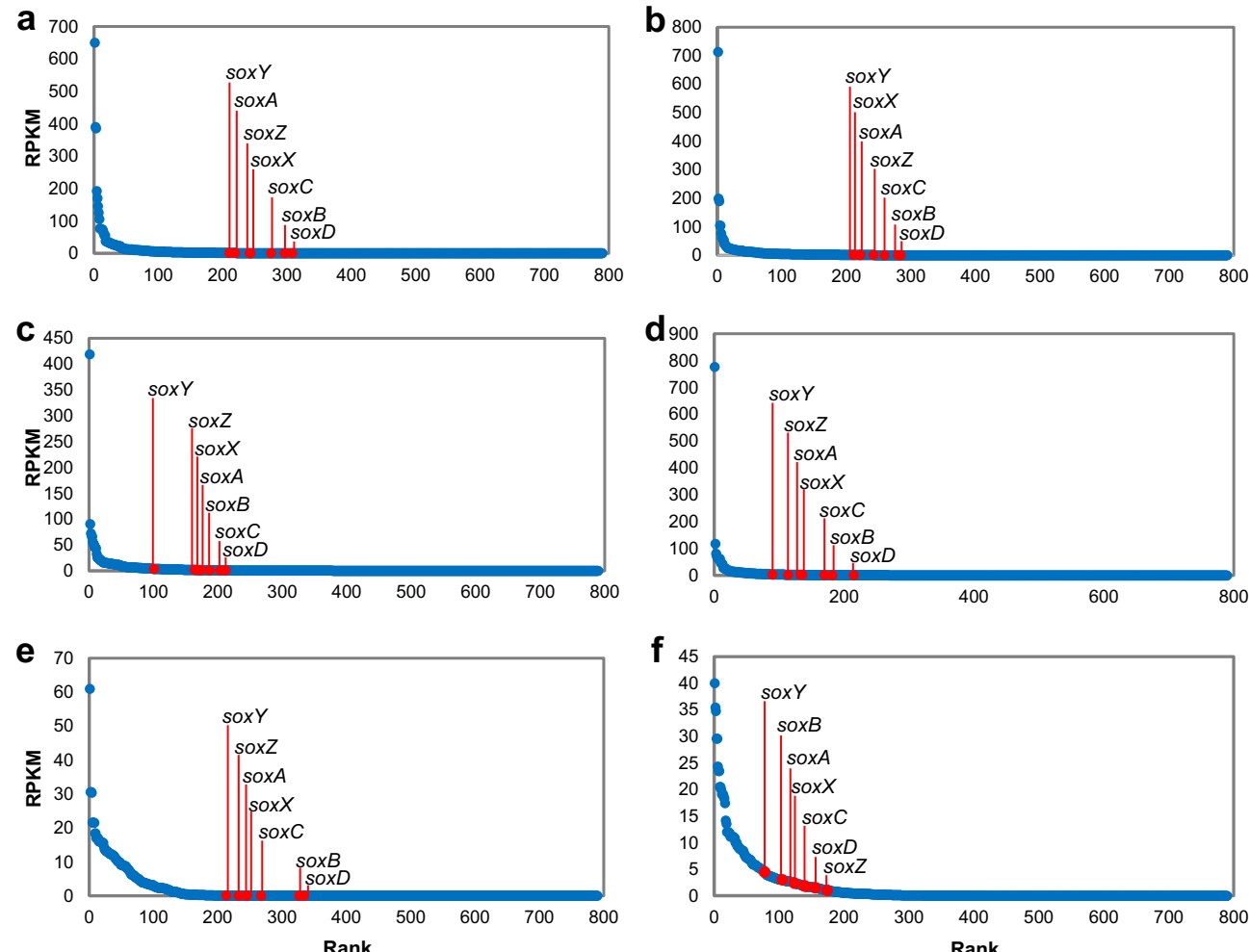

**Fig. 3 | Expression of *sox* genes in the six biofilm metatranscriptomes.** The biofilms were collected in Sep 2020 (**a**), Nov 2020 (**b**), Jan 2021 (**c**), Mar 2021 (**d**), May 2021 (**e**), and Jul 2021 (**f**). Genes were predicted from assembled metatranscriptomes and annotated by KEGG. Rank analyses of the seven *sox* genes among all the energy metabolism KEGG genes are based on their Reads Per Kilobase per Million mapped reads (RPKM) values. Source data are provided as a Source data file.

To estimate the contribution of the *Roseobacter* group to thiosulfate oxidation in marine biofilms, we analyzed the taxonomic affiliations of *sox* genes in the six biofilm metagenomes sequenced in the present study. Assembly of the six metagenomes generated 5,698,001, 5,046,598, 6,876,380, 2,461,921, 5,467,731, and 8,910,019 contigs, respectively, predicted to represent 3,182,941, 2,784,653, 3,334,736, 1,286,705, 2,810,401, and 6,312,794 close-ended ORFs (Supplementary Data 3). Gene annotation and taxonomic analysis of *soxX* and *soxA* ORFs revealed that a large percentage of these ORFs were affiliated to *Roseobacter*, such as *Sulfitobacter*, *Ruegeria*, *Roseovarius*, *Octadecabacter*, *Phaeobacter*, and *Roseibium* (Supplementary Fig. 12). Taxonomic analyses of two anaerobic respiration-related genes, *napA* and *nirK*, were also performed. While only a few *napA* ORFs belonged to the *Roseobacter* group, a substantial number of *nirK* ORFs were affiliated to *Roseobacter*-affiliated genera, including *Sulfitobacter*, *Phaeobacter*, *Roseovarius*, and *Roseobacter* (Supplementary Fig. 13).

### In situ metatranscriptomic profiling of *Roseobacter*-affiliated *sox* genes

To study the in situ activity of biofilm *Roseobacter* strains, metatranscriptomic sequencing was performed on the six coastal marine biofilms collected in the present study. Assembly of the six metatranscriptomes generated 2,515,854, 3,111,162, 3,316,686, 3,845,996, 855,508, and 1,124,063 contigs, respectively, predicted to represent 2,999,230, 3,703,787, 1,242,031, 1,537,969, 402,727, and 1,189,877 close-ended ORFs (Supplementary Data 4). After read mapping and ORF annotation, the transcription levels (indicated by Reads Per Kilobase per Million mapped reads or RPKM) of all the genes involved in energy metabolism were compared and, in all cases, all the *sox* genes were ranked within the top 50% (Fig. 3a–f for biofilm_1, _2, _3, _4, _5, _6, respectively). Among the seven *sox* genes, *soxY* tended to be more active than the other genes. For example, in the biofilm collected in Jul 2021 (biofilm_6), *soxY* ranked in the top 10% of all the genes involved in energy metabolism (Fig. 3f).

To study the proportion of active transcripts in the isolated biofilm *Roseobacter* strains, the metatranscriptomic reads were mapped to the 54 genomes. In total, these genomes recruited 3.19%, 4.00%, 3.79%, 3.80%, 3.85%, and 10.35% reads from the six metatranscriptomes, respectively (Supplementary Fig. 14), suggesting that these bacteria are active in the biofilm communities. To further estimate the contribution of *Roseobacter* strains to thiosulfate oxidation, *soxA* and *soxX* ORFs (as these two genes have key functions in thiosulfate oxidation) were extracted from assembled biofilm metatranscriptomes and their taxonomic affiliations were analyzed. At the genus level, up to 21.05% of the *soxA* and up to 20.00% of the *soxX* genes (Supplementary Fig. 15) were associated with *Sulfitobacter*, and these two genes also accounted for relatively high percentages in other *Roseobacter*-affiliated genera, such as *Roseobacter*, *Roseovarius*, and

*Roseibium*. These results suggested that the *Roseobacter* group is the major contributors to thiosulfate oxidation in marine biofilms. In addition, taxonomic analysis of *napA* and *nirK* was performed. While none of the top 20 *napA* ORFs belonged to the *Roseobacter* group, a substantial number of *nirK* ORFs belonged to *Roseobacter*-affiliated genera, such as *Sulfitobacter*, *Roseovarius*, *Roseobacter*, and *Phaeobacter* (Supplementary Fig. 16), suggesting that nitrite might be used as electron acceptor by the *Roseobacter* group in the biofilms.

### Experimental evidence for anaerobic thiosulfate oxidation

Before conducting thiosulfate oxidation experiments for representative strains of the biofilm *Roseobacter* strains, we explored their general physiological characteristics. Phenotypes of nine representative strains (nine strains representing the nine different genera) were observed using transmission electron microscopy (TEM). All the observed strains displayed rod or oval shapes with cell lengths ranging from 0.5 to 3.0 µm (Supplementary Fig. 17). When cultured in the complex medium marine broth 2216, all the strains could grow aerobically and anaerobically with an optional growth temperature of 22–26 °C, and several strains accumulated comparable biomass when grown in these two conditions (Supplementary Fig. 18). For example, the maximum optical densities at 600 nm ($OD_{600}$) of *P. inhibens* M623 reached 1.28 when grown aerobically, while reaching 1.20 when grown anaerobically (Supplementary Fig. 18).

Thiosulfate oxidation was examined in the nine representative strains under both aerobic and anaerobic conditions. These strains were grown planktonically in artificial seawater media (i.e., minimum media) with 10 mM thiosulfate. The accumulation of sulfate (>0.1 mM) in the cultures was observed for the six strains (M382, M583, M619, S051, S190, and S2214) possessing the *sox* genes by barium precipitation after removing bacterial cells from the cultures, and sulfate production was detected (Supplementary Fig. 19). In particular, the sulfate production rate of Rhodobacteraceae sp. M382 is shown (Fig. 4a). This strain was selected as a model for the following analyses and experiments, because it probably represents a new genus and its genetic manipulation can be achieved. In both conditions, the sulfate concentration in M382 cultures increased along with the bacterial growth (Fig. 4a, b). In contrast, the sulfate concentrations in the cultures of two mutant strains of M382, M382-*ΔsoxX*, and M382-*ΔsoxA*, showed a slight decrease (Fig. 4a, b), confirming the role of *sox* genes in thiosulfate oxidation and sulfate production. The slight decrease in sulfate concentration may be due to assimilatory consumption of sulfate originally present in seawater. To confirm the production of barium sulfate, the pellet of a tested sample was examined by scanning electron microscopy (SEM) (Fig. 4c), and the dominant spectra of barium, sulfur, and oxygen were observed (Fig. 4d). These results suggested that the biofilm *Roseobacter* strains use *sox* genes for thiosulfate oxidation under both aerobic and anaerobic conditions. These results also showed that when grown planktonically, mutation of the *soxX* or *soxA* genes did not affect the growth of M382 (Fig. 4a, b), and motivated us to examine the growth of wild-type M382, M382-*ΔsoxX*, and M382-*ΔsoxA* in the biofilm state. After culture under aerobic or anaerobic conditions in artificial seawater media with 10 mM thiosulfate in biofilms, the wild-type strain displayed a significantly higher cell density than the two mutants (Supplementary Fig. 20), suggesting that thiosulfate oxidation may affect the bacterial growth when they have formed biofilms rather than grown planktonically.

### Transcriptomics and membrane proteomics supporting anaerobic thiosulfate respiration

We next investigated the thiosulfate metabolism strategy in M382 biofilm using transcriptomics. Gene transcription profiles of M382 biofilms cultured in complex media with or without thiosulfate were studied (data are shown in Supplementary Data 5). In total, 68 KEGG genes (genes that could be annotated by KEGG) were significantly

(fold-change >2 and *P* value <0.05 by Student's *t* test) up-regulated by thiosulfate, 58 KEGG genes were down-regulated, while 4607 KEGG genes remained unchanged (Supplementary Fig. 21). Notably, the transcription of all *sox* genes was induced by thiosulfate (Fig. 5a). For example, there was over 14-fold induction of *soxC* transcription (Fig. 5a). Moreover, several genes related to anaerobic respiration were also induced by thiosulfate, including the ferredoxin-type protein-encoding gene *napH* (K02574), D-lactate dehydrogenase *dld* (K03777), the cytochrome c biogenetic gene *ccdA* (K06196), and anaerobic dimethyl sulfoxide reductase *dmsC* (K07308) (Fig. 5b). In addition, two ORFs encoding PorD (K11069), the substrate-binding protein of the spermidine/putrescine transport system, were up-regulated by thiosulfate, as well as an ORF encoding the outer membrane protein OmpW (K07275) (Fig. 5c), which has been reported to play a role in biofilm formation[44,45]. As examples, PotD is a spermidine/putrescine-binding periplasmic protein, and its over-expression strongly promotes biofilm formation in *Escherichia coli*[44], and the mutant strain *ΔompW* of *Acinetobacter baumannii* showed reduced biofilm formation[45]. The full list of the genes with significantly higher transcription levels in the thiosulfate culture are shown in Supplementary Fig. 22, while the down-regulated genes are shown in Supplementary Fig. 23. It is worth mentioning that transcription of the *sat* gene (K00958) and three *cys* genes (K00381 and two K00390 genes) were down-regulated by thiosulfate (Supplementary Fig. 23), suggesting the inhibitory effect of thiosulfate on sulfate reduction.

Because many of the thiosulfate-induced genes in the M382 biofilm were likely to encode proteins (e.g., OmpW as mentioned above) that are localized in the cell membrane, we used proteomics to explore the rearrangement of membrane proteins caused by culturing with thiosulfate, and the final results supported our expectation. Up to 304 proteins showed significantly (Student's *t*-test, *P*-value <0.05) increased abundance in the cell membrane after thiosulfate treatment, while the abundance of 149 proteins decreased (Supplementary Fig. 24 and Supplementary Data 6). It was noteworthy that the seven SOX proteins displayed 2.40-64.93 fold changes in the membrane proteome after addition of thiosulfate (Fig. 5d and Supplementary Data 6), suggesting their cell-membrane localization during thiosulfate oxidation. Moreover, consistent with the transcriptomic data, thiosulfate elevated the levels of several proteins involved in anaerobic respiration (Fig. 5e and Supplementary Data 6) in the cell membrane. For example, lactate dehydrogenase is an important enzyme in the anaerobic metabolic pathway where it catalyzes the reversible conversion of lactate to pyruvate[46], and cytochrome c peroxidase (CCP) has been observed to degrade hydrogen peroxide in anoxic *E. coli*, with transcription of *ccp* requiring the absence of oxygen[47]. In addition, proteins responsible for biofilm formation showed higher abundance in the thiosulfate-treated biofilms (Fig. 5f and Supplementary Data 6). For example, in Gram-negative bacteria, the general secretory pathway includes the type II secretion system and the type IV pilus[48], both of which play important roles in biofilm formation in a variety of species[49,50]. Based on the increased abundance of proteins involved in anaerobic respiration and biofilm formation, we speculated that thiosulfate is used as an energy source in the biofilm state. To test this speculation, we measured the proton motive forces (PMFs) in the M382 biofilms with or without thiosulfate. This showed that the addition of thiosulfate increased PMF production (Supplementary Fig. 25). Finally, a schematic model showing the promotion of anaerobic respiration and biofilm formation by thiosulfate oxidation is provided (Fig. 6).

## Discussion

### Novel ecological implications for marine *Roseobacter* group

Although most of the known members of this clade are culturable, *Roseobacter* strains are important and require further in-depth

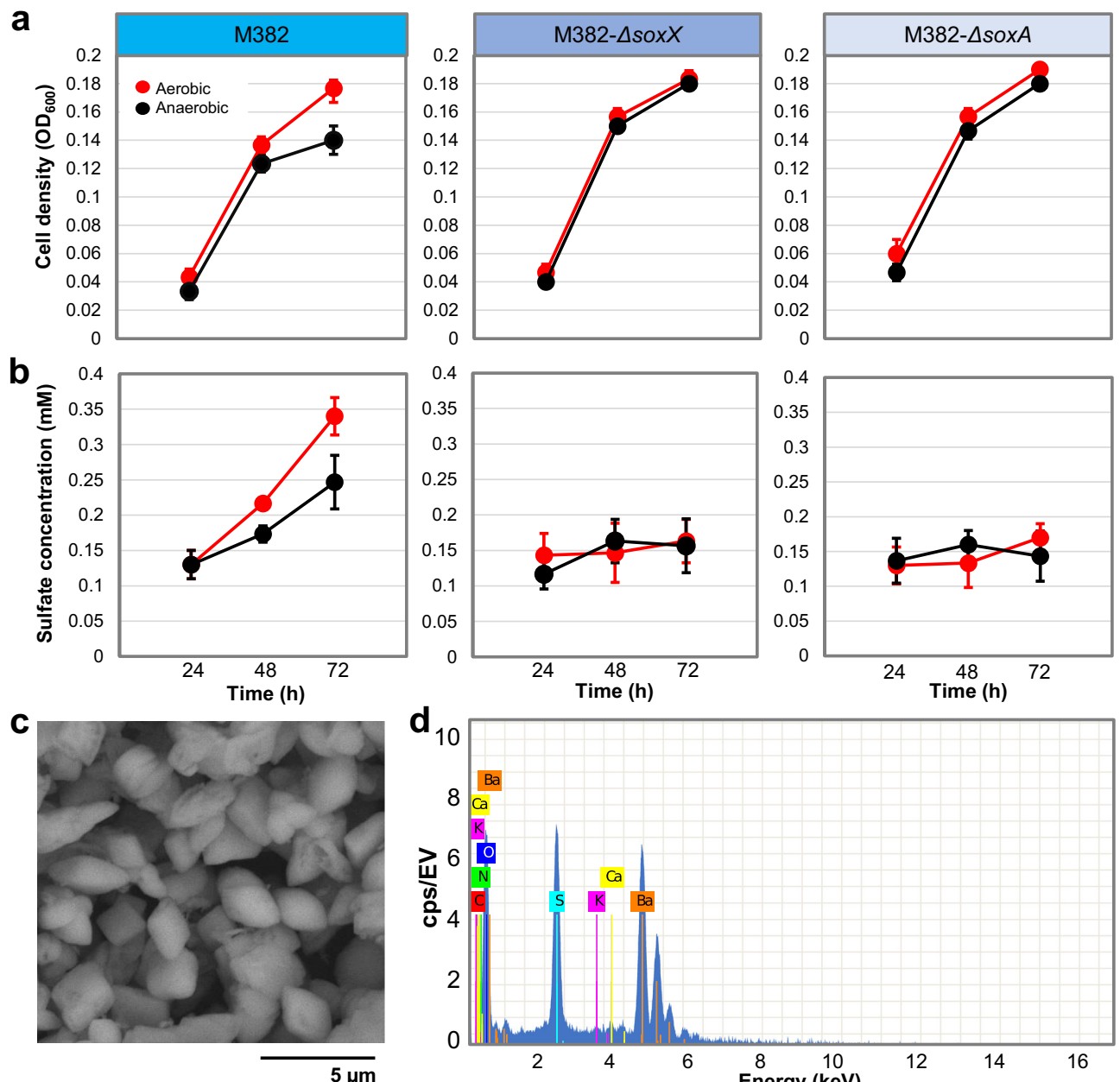

**Fig. 4 | Experimental study of thiosulfate oxidation by Rhodobacteraceae sp. M382.** The wild-type M382 and its two mutant strains were grown planktonically in minimum media with 10 mM thiosulfate, followed by measurement of optical densities of the cells at 600 nm (**a**) and the sulfate concentrations in the media (**b**) at three time points. Values are shown as mean ± s.d. (n = 3 biologically independent replicates). **c** Scanning electron micrograph of the pellet (barium sulfate) produced by adding barium chloride to the diluted culture of M382 grown with thiosulfate. **d** Elemental spectra of the pellet. Source data are provided as a Source data file.

investigation due to the diversity of their habitats and metabolic activities[51]. Here, we performed a comprehensive investigation of biofilm *Roseobacter* strains, focusing specifically on their roles in thiosulfate oxidation. The results from functional genomics, global metagenomics, and metatranscriptomics enhance the ecological view of the marine *Roseobacter* group from three major perspectives.

First, the discovery of novel biofilm plasmids in *Roseobacter* strains. There is experimental evidence that biofilm is a hot spot of gene exchange mediated by the transfer of mobile elements including plasmids[52], yet the function and diversity of plasmids in marine *Roseobacter* strains remains largely unknown. Here, we found that consistent with a previously-reported plasmid that encodes a rhamnose operon in *Phaeobacter inhibens* DSM 17395[31], the biofilm

*Roseobacter* strains contain a variety of plasmids that encode genes for the biosynthesis of saccharides, including succinoglycan, capsular polysaccharide, adhesin, and mannose, which facilitate adhesion during biofilm formation. Considering that the copy number of biofilm-related genes in plasmids is probably higher than those in chromosomes, our results highlight the important role of plasmids in carrying niche-specific traits, and that biofilm plasmids are more diverse than previously expected. Moreover, the presence of nitrogen respiration genes in biofilm plasmids of *Roseobacter* strains suggests their roles in facilitating electron transfer, which might be important for thiosulfate oxidation.

Second, niche partitioning of marine *Roseobacter* strains is observed consistently from surface waters to the deep ocean. Previous

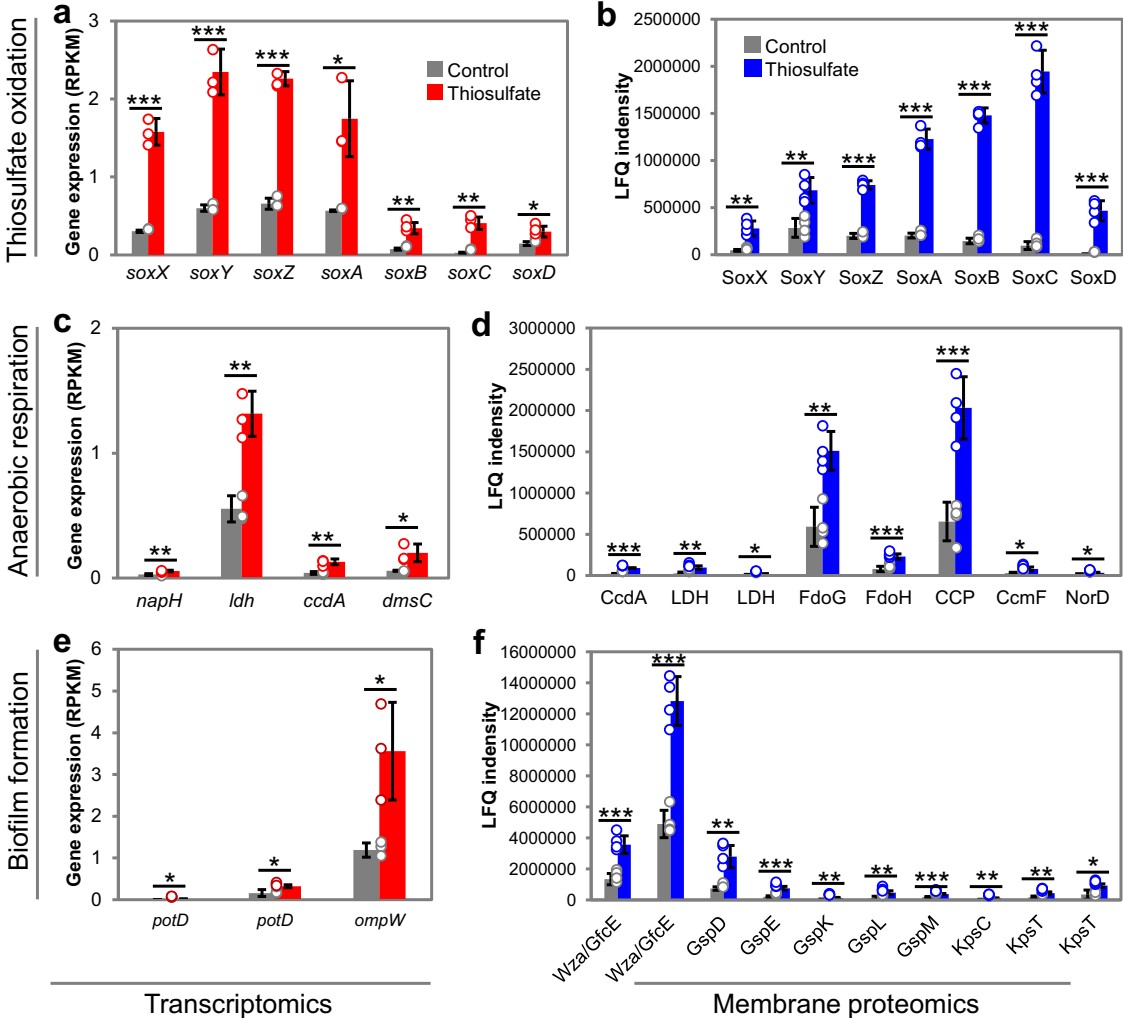

**Fig. 5 | Transcriptomics and membrane proteomics analyses of the M382 biofilm in response to thiosulfate.** Genes or proteins associated with thiosulfate oxidation (**a** transcriptomics; **b** membrane proteomics), anaerobic respiration (**c** transcriptomics; **d** membrane proteomics), and biofilm formation (**e** transcriptomics; **f** membrane proteomics) are shown. Full names of genes in the transcriptomics results: *napH*, ferredoxin-type protein; *ldh*, lactate dehydrogenase; *ccdA*, cytochrome c-type biogenesis protein; *dmsC*, anaerobic dimethyl sulfoxide reductase subunit C; *potD*, spermidine/putrescine transport system substrate-binding protein; *ompW*, an outer membrane protein. Full names of proteins in the proteomics results: CcdA, cytochrome c-type biogenesis protein; Ldh, lactate dehydrogenase; FdoG, formate dehydrogenase major subunit; FdoH, formate dehydrogenase iron-sulfur subunit; CCP, cytochrome c peroxidase; CcmF, cytochrome c-type biogenesis protein; NorD, nitric oxide reductase. Wza or GfcE,

polysaccharide biosynthesis/export protein; GspDEKLM, general secretion pathway proteins; KpsC, capsular polysaccharide export protein; KpsT, capsular polysaccharide transport system ATP-binding protein. In the bar charts, values are shown as mean ± s.d. (*n* = 3 biologically independent replicates for transcriptomics and *n* = 4 biologically independent replicates for proteomics). Statistics in both analyses were performed using two-sided Student's *t* test with thresholds of fold change >2 and *P* value <0.05 (\**P* value <0.05; \*\**P* value <0.01; \*\*\**P* value <0.001). Exact *P* values in **a**: 0.0002, 0.0005, 1.7E-05, 0.0136, 0.0031, 0.0013, 0.0225; **b**: 0.0011, 0.0032, 7.5E-07, 1.5E-06, 7.8E-08, 3.8E-06, 0.0001; **c**: 0.0067, 0.0032, 0.0039, 0.0251; **d**: 4.5E-05, 0.0014, 0.0173, 0.0015, 0.0006, 0.0008, 0.0183, 0.0120; **e**: 0.0469, 0.0353, 0.0255; **f**: 0.0006, 0.0001, 0.0013, 0.0002, 0.0053, 0.0075, 0.0004, 0.0048, 0.0023, 0.0112. Source data are provided as a Source data file.

studies have reported that *Roseobacter* distribution was associated with a coccolithophore bloom in the Northwestern Black Sea and was strongly dependent on dimethylsulfoniopropionate-producing phytoplankton species[53,54]. Here, we provide a comprehensive picture of the selective distribution of *Roseobacter* strains between biofilm and water, suggesting that biofilm formation contributes significantly to *Roseobacter* distribution in the global ocean. Considering that the biofilm and seawater metagenomes were collected across different seasons, the enrichment of these *Roseobacter* strains in marine biofilms is not majorly affected by bloom conditions. Instead, the preference for the biofilm niche may be driven by redox stratification, given that these *Roseobacter* strains are capable of growing in both aerobic and anaerobic conditions.

Third, *Roseobacter* strains are likely to be one of the major thiosulfate consumers in hydrothermal vent environments. In a previous study[14], *Roseobacter* strains were isolated from hydrothermal vent environments, such as TB66, isolated from a Galapagos vent sample. *Roseovarius tolerans* is closely associated with TB66 and shows thiosulfate-oxidizing activity[14]. Many of the reported *Roseobacter* strains in vent fields inhabit biofilm mats and it is assumed that a *Sulfitobacter* strain can fix carbon dioxide while using reduced sulfur compounds for growth[55]. Thus, *Roseobacter* strains, particularly those that inhabit biofilms, may have been neglected during the profiling of sulfur cycling in hydrothermal vent ecosystems in previous studies that have described Campylobacterota (previously named Epsilonproteobacteria) as the major players involved in thiosulfate oxidation[36,56].

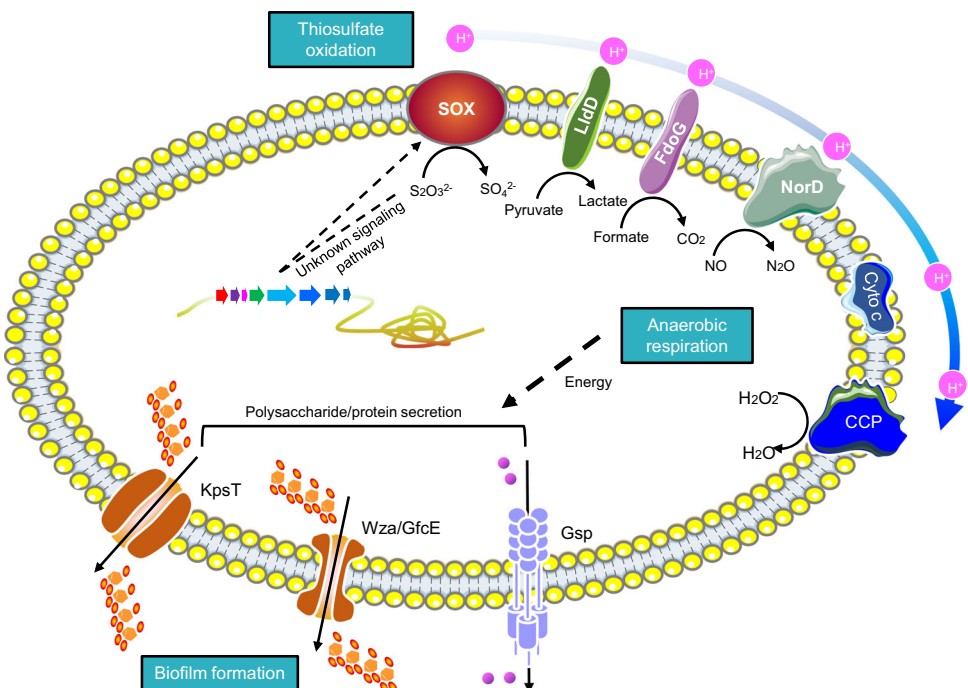

**Fig. 6 | A schematic model to show the anaerobic respiration and biofilm formation driven by thiosulfate oxidation.** The model is majorly based on the cell membrane proteomics results and the functions of proteins with increased abundance are summarized.

## Anaerobic respiration contributes significantly to marine thiosulfate oxidation

The presence of thiosulfate oxidation has been extensively documented in symbionts of deep-sea animals[57,58]. Due to a lack of experimental evidence, however, the precise mechanisms of thiosulfate oxidation in these symbionts are not clear. However, it can be speculated that, in these bacteria, thiosulfate oxidation may occur anaerobically as oxygen is scarce in many deep-sea environments and its concentration in animal tissues is probably much lower than in the surrounding seawater. Similarly, microenvironments at the base of biofilms are largely anaerobic due to the accumulation of polymeric substances that prevent oxygen penetration[59–61]. Here, we have provided consistent evidence that a large percentage of expressed *sox* genes in biofilms are associated with the *Roseobacter* group, and the activation of genes involved in anaerobic respiration suggests that thiosulfate oxidation is mostly anaerobic. The activation of *Roseobacter*-affiliated *nirK* genes in biofilm metagenomes suggested the utilization of nitrite as the final electron acceptor. Therefore, although there are no data to quantify the contribution of symbiont and biofilm-associated microbes in marine thiosulfate oxidation, it can be concluded that a large proportion of thiosulfate in marine environments is probably oxidized anaerobically.

## Thiosulfate oxidation and biofilm formation

Despite numerous studies that have investigated the process of biofilm development, the energy that promotes biofilm formation in marine environments is poorly understood. It is suggested that biofilm formation requires energy in addition to that supporting the basic life of microbes[62]. Here, we provide proteomic evidence that thiosulfate can change the cell membrane protein composition of a *Roseobacter* strain to enhance biofilm formation and anaerobic respiration. Consistently, mutation of the *sox* genes reduces cell density in biofilms and addition of thiosulfate promotes PMF production, suggesting that thiosulfate oxidation is likely to contribute to energy production. In addition, in other environments, thiosulfate oxidation is known to be coupled to several important physiological activities (e.g., in symbionts, thiosulfate oxidation is believed to be an essential energy source of carbon

fixation[57,58]). Therefore, we propose that thiosulfate is likely to provide energy for biofilm formation. However, this notion needs to be further demonstrated in future studies.

## Conclusion and perspective

To answer the two key questions raised at this paper's beginning, we have shown that (1) *Roseobacter* strains are the major players in thiosulfate oxidation in marine biofilms, and (2) thiosulfate in marine biofilms is largely oxidized anaerobically and is of great significance to the surface-associated bacteria. Limitations exist for the current study. For example, here we have focused on culturable *Roseobacter* strains in marine biofilms while metagenome-assembled genomes were not used. This is largely due to the high microbial diversity in marine biofilms, which hinders the recovery of high-quality genomes from biofilm metagenomes. Moreover, here we focused on the analyses of *sox* system in thiosulfate oxidation, which might be mediated by other genes (e.g., the thiosulfate dehydrogenase *tsdA* that works together with *soxB*[63]). Nevertheless, our results revealed the identity and functional properties of surface-associated *Roseobacter* strains that may contribute to the key function of thiosulfate oxidation in marine ecosystems. Considering the complexity of the natural sulfur cycle, the previously neglected role of surface-associated bacteria in the marine sulfur cycle requires further clarification.

## Methods

### Biofilm sampling and bacterial isolation

Biofilms for bacterial isolation were collected in March 2019 in the coastal area of Qingdao, China (36.05, 120.43). Natural biofilms on rock surfaces at a depth of 1–2 m in the subtidal zone were scraped off using sterile cotton tips and immediately transferred to the laboratory. The cotton tips were thoroughly rinsed with 10 ml of 0.1-μm-filtered and autoclaved seawater before being diluted 10 and 100 times. For each biofilm, 100 μl aliquots were spread on marine agar 2216 plates (BD, Difco). Subsequent microbial incubations were conducted under aerobic conditions at 25 °C with artificial sunlight radiation. Colonies were examined under a dissecting microscope for morphological characteristics including size, color, shape, and surface topography.

Conspicuous colony types were isolated, and DNA was extracted from the cells by heating at 100 °C for 5 min. PCR was conducted to amplify the 16S rRNA genes using the universal primers 27F (5'-AGAGTTT-GATCCTGGCTCAG-3') and 1492R (5'-GGTTACCTTGTTACGACTT-3'), followed by Sanger sequencing (for each strain, both forward and reverse reactions were performed to obtain the full-length 16S rRNA gene sequence) at the Beijing Genomics Institute (BGI, China) to identify the taxonomy of the isolates.

## Genome sequencing and analyses

Prior to genome sequencing, the DNA of cultured bacterial strains was extracted using the TIANamp Genomic DNA Kit (Tiangen Biotech, China), following the manufacturer's protocol with modification. Lysozyme was added to a final concentration of 10 mg/ml and incubated at 37 °C for 30 min. The lysis buffer (buffer GA), and buffers GB, GD, and GW were added as specified in the manufacturer's protocol. DNA integrity was examined by agarose gel electrophoresis in a Mini-Sub Cell GT System (Bio-Rad, USA). PacBio and Illumina sequencing was performed at the Novogene Bioinformatics Institute (Novogene, China). For each bacterial strain, PacBio single-molecular real-time (SMRT) sequencing was performed using the circular consensus sequencing (CCS) strategy to generate 1 Gb of data (N50 read length >9000 bp), and Illumina sequencing was performed on the NovaSeq 6000 system to generate 2 Gb of data (read length = 150 bp). Preliminary assembly and correction were performed with SMRT Link v5.0.1 (CCS = 3, minimum accuracy >0.99), and then corrected by the Illumina data using Minimap2 (Minimap2: pairwise alignment for nucleotide sequences). In detail, the PacBio-derived contigs were mapped with the Illumina reads, and the locations without alignment were corrected according to the contigs assembled from Illumina reads. The chromosome and plasmid sequences were distinguished based on the reads coverage and screened by BLASTn to form a complete genome. Genome annotation was performed on a local Linux platform. Ribosomal RNA genes (rRNA) and transfer RNA genes (tRNA) were predicted using the software Barrnap (https://github.com/tseemann/barrnap) and Aragorn[64], respectively. ORFs were predicted using Prodigal (version 2.60)[65] in a single-genome analysis model with close-end ORF prediction. For phylogenetic analysis, 31 essential marker genes (*dnaG, frr, infC, nusA, pgk, pyrG, rplA, rplB, rplC, rplD, rplE, rplF, rplK, rplL, rplM, rplN, rplP, rplS, rplT, rpmA, rpoB, rpsB, rpsC, rpsE, rpsI, rpsJ, rpsK, rpsM, rpsS, smpB*, and *tsf*) were extracted from the genomes using AMPHORA2[66]. The protein sequences of these marker genes were aligned using MEGA (version 6.05)[67] to construct a maximum-likelihood tree under the Jones–Taylor–Thornton mode. The marker genes were aligned individually and then concatenated to generate alignments for tree construction. The tree was built in MEGA[67] and bootstrap values were calculated with 500 replicates. ANI was calculated using the software fastANI[68] installed in a local Linux system. Two different species show <95% ANI[69], and two different strains have <99% ANI[70]. For functional gene annotation, the protein sequences of a given genome searched with BLASTp (E-value <1e-7) against the KEGG database (2022 version). Metabolic pathways were reconstructed using the online server KEGG Mapper (https://www.genome.jp/kegg/mapper.html).

## Metagenomic sequencing and analyses

Six new biofilms were collected in the present study, followed by metagenomic and metatranscriptomic sequencing. These biofilms were scraped from subtidal stone surfaces in the coastal area (36.05, 120.43) of Qingdao, China, at six time points (Sep 2020, Nov 2020, Jan 2021, Mar 2021, May 2021, and Jul 2021). DNA extraction was performed using the TIANamp Genomic DNA Kit (Tiangen Biotech, China) and the detailed steps are the same as those described in the "Genome sequencing and analyses" section. Metagenomic sequencing was performed on the NovaSeq 6000 system at the Novogene Bioinformatics

Institute (Beijing, China). Paired-end reads with a read length of 150 bp were generated after construction of libraries with 350 bp insertion. Metagenomic analysis was conducted following the procedures used in our previous studies[18,71]. In detail, quality control of the Illumina sequences was performed on our local server using the software NGS QC Toolkit (version 2.0)[72]. Reads containing adapters, low-quality (quality score <20) reads, or unpaired high-quality reads were removed. Metagenomes published in previous studies[18,36–43] were also downloaded from the NCBI SRA database using the fastq-dump script in a Linux system. To calculate the relative abundance of the isolated strains in biofilm-associated and free-living metagenomes, all the metagenomes included for analyses were normalized to 1,000,000 reads, and all the reads were trimmed to 101 bp. The metagenome reads were mapped to bacterial genomes using BBMap[73] (minimum alignment identity = 0.80). Relative abundance was calculated by counting the numbers of mapped reads. To determine the taxonomic affiliation of a given gene in the six biofilm metagenomes sequenced in the present study, the clean metagenomic reads were assembled using MEGAHIT[74], followed by ORF prediction using Prodigal in a Meta mode, returning only close-ended ORFs. The ORFs were annotated by DIAMOND[75] BLASTp (E-value <1e-7) searching against the KEGG database (2022 version). Taxonomic affiliation at the genus levels was profiled using the 2022 Linux version of Kaiju[76], with the kaiju_db (2022 version) as the reference.

## Metatranscriptomic sequencing and analyses

The six biofilm samples (sampling locations and collection times described above) were immediately transferred to the laboratory, frozen using liquid nitrogen, and stored at -80 °C. RNA extraction was performed using a TRIzol lysis method and Ribo-Zero Strand-specific libraries were constructed using the NEBNext Ultra RNA Library Prep Kit (NEB, USA). Following the manufacturer's recommendations, biotin-labeled oligonucleotides complementary to rRNA or other non-coding RNAs were mixed with the total RNA, and mRNA was selectively retained and converted to cDNA for library preparation. The libraries (insert length = 350 bp) were sequenced on the Novaseq 6000 system at the Novogene Bioinformatics Institute (Beijing, China) to generate 60 Gb of data (paired-end reads with a length of 150 bp) for each biofilm. Clean reads were obtained using the NGS QC Toolkit (version 2.0)[72]. To determine the percentage of the *Roseobacter* strains, the metatranscriptomic reads were mapped to the 54 genomes using BBMap[73] (minimum alignment identity = 0.80) and the relative abundance was calculated by counting the numbers of mapped reads. To determine the expression level and taxonomic affiliation of a given gene in the metatranscriptomes, the metatranscriptomic reads were assembled using MEGAHIT[74]. ORFs were predicted from the assembled contigs using Prodigal in a Meta mode, and only close-ended ORFs were returned for further analysis. The ORFs were annotated by DIAMOND[75] BLASTp (E-value <1e-7) searching against the KEGG database (2022 version). The clean metatranscriptomic reads were mapped to ORF sequences using Bowtie2 v2.4.2[77]. The coverage of a given gene (e.g., *soxA*) was calculated using SAMtools v1.11[78] to determine gene expression profile, displayed in RPKM. Taxonomic affiliation at the family and genus levels was profiled using Kaiju[76], with the kaiju_db (2022 version) as the reference.

## Gene knockout

In-frame gene deletion was performed based on homologous recombination using the pHG1.0 plasmid hosted by *E. coli* DB3.1. PCR primers were designed to amplify two regions flanking the upstream and downstream regions of the target gene. In this study, two genes were knocked out using the primers for *soxX* Up-F (GGGGACAAGTTTGTA-CAAAAAAGCAGGCTTGGATTTCACTGATGGCGGG), *soxX* Up-R (AAGT GC-GCCTAATCGCGTAGGCGGCCAATGTCAGAGATGT), *soxX* Down-F (CTACGCGATTAGGCGCACTTTG-ACCGCTCAGGAAATCGAAG), *soxX*

Down-R (GGGGACCACTTTGTACAAGAAAGCTGGGTCATGTC-GAGAA CCGACTGGC), and for *soxA* Up-F (GGGGACAAGTTTGTACAAAAAAG CAGGCTATCTTTGTC-CCGCCTTCGC), *soxA* Up-R (AAGTGCGCCTAAT CGCGTAGAGATCGGCGCTTTCGTCG), *soxA* Down-F (CTACGCGAT-TAGGCGCACTTTGTATGTTGCATCCCGTGGC), and *soxA* Down-R (GG GGACCACTTT-GTACAAGAAAGCTGGGTTTGAAAAGCTCGGCGGGT TC). The two flanking regions were fused via a complementary linker region that was added to the 5′ end of the primers Up-R and Down-F. The primers Up-F and Down-R also contained recombination sequences for ligation with the pHG1.0 plasmid. Then, using the primers Up-F and Down-R and a Hi-Fi Taq DNA Polymerase kit (Invitrogen, Waltham, MA, USA), a "mutated copy" of the target gene was generated by PCR. After checking the relative concentrations of the PCR products and the extracted pHG1.0 plasmid using a Nanodrop spectrophotometer (Thermo Fisher, Waltham, MA, USA), the PCR product was inserted into pHG1.0 using Gateway BP clonase II enzyme mix (Thermo Fisher). Recombination was performed for two hours at 25 °C, according to the manufacturer's instructions. The ligated DNA product was then transformed into *E. coli* wm3064 competent cells using the heat-shock method (42 °C for 30 s) and plated on LB agar with 50 μg/ml of gentamicin and 50 μg/ml 2,6-Diaminopimelic acid (DAP) and incubated for 16 h at 37 °C. The successfully transformed cells were selected using PCR. Then, DNA conjugation between *E. coli* wm3064 carrying the "mutated copy" DNA fragment in pHG1.0 and strain M382 were performed using the following steps: (1) wm3064 and M382 were grown in LB media with 50 μg/ml of gentamicin and 50 μg/ml of DAP and marine broth 2216, respectively, to an $OD_{550}$ value of about 0.4 at 37 °C and 25 °C, respectively; (2) The donor cells (2 ml of wm3064) and recipient cells (1 ml of M382) were mixed and precipitated by centrifugation at $1792 \times g$ before resuspension in 200 μl of marine broth 2216 and plated on marine broth 2216 media with 50 μg/ml gentamicin. As wm3064 cannot grow without DAP, this step allows the selection of M382 with the "mutated copy" DNA fragments; (3) After overnight incubation at 25 °C, approximately 500 colonies were selected and screened by PCR for successful recombination; (4) The candidate cells were then transferred to marine broth 2216 media with 20% sucrose to obtain colonies with two-step recombination.

### Bacterial growth experiment and thiosulfate oxidation measurement

During the bacterial growth experiment, bacteria were cultured on marine broth 2216 media (BD, Difco) at 25 °C. For aerobic growth, the strains were cultured in air, while for anaerobic growth, the strains were cultured in a mix of hydrogen gas, carbon dioxide, and nitrogen gas (5/5/95%). The maximum biomass was recorded by growing the bacterial strains for over seven days and measuring the $OD_{600}$ values every 24 h. The growth experiments were repeated three times on three different occasions. The phenotypes of cultured strains were observed using a JEM-1200EX TEM system (JEOL, Japan).

During the thiosulfate oxidation experiment, bacteria were grown planktonically in artificial seawater media containing 5% of sea salts, 10 mM thiosulfate, 10 mM glucose, 10 mM of HEPES (pH 8.0), 200 μM of sodium hydrogen phosphate, 500 μM of ferric chloride, and 10 mM of ammonium chloride. Aerobic and anaerobic conditions were set as described above. The sulfate concentrations in the media were measured using a sulfate assay kit (BioVision Inc Milpitas, CA, USA). Specifically, cells cultured for 72 h were centrifuged at $18,928 \times g$ for 10 minutes and 20 μl of the supernatant was transferred to 190 μl of the treatment solution provided by the kit. Then, 100 μl of the detection buffer was added and allowed to react for 15 min at room temperature. Standards were prepared by adding 0, 50, 100, and 200 μl of the standard solution to 100 μl of the detection buffer. Different volumes of the treatment solution were added to the standards to make up equal volumes. OD values were measured at 600 nm and the sulfate concentrations in the samples were determined using the formula

sulfate (mM) = $OD_{600}$ sample/standard concentration slope × 10. The barium sulfate pellets were confirmed using a TESCAN VEGA3 SEM system (TESCAN, Czechoslovakia). For the wild-type M382 and its two mutants, thiosulfate accumulations after planktonic cultivations of 24, 48, and 72 h were measured. In addition, to study the influence of *sox* gene mutation on M382 growth in the biofilm state, biofilms of the wild-type strain and the two mutants were grown on the bottom of six-well plates in artificial seawater media with 10 mM thiosulfate. After 72 hours, bacterial cells were scattered by "blowing and sucking" using a 1-ml pipette, harvested by centrifugation, and re-suspended in equal volume of media before measured at $OD_{600}$.

### Transcriptomics

M382 strains were grown in triplicate cultures in marine broth 2216 with or without 10 mM thiosulfate and incubated in six-well plates at 25 °C for 48 h. After removal of the liquid with suspended bacteria, the biofilms formed on the bottom of the wells were washed with fresh media before transfer to Falcon tubes. The cells were harvested at $5000 \times g$ for five minutes and immediately transferred to liquid nitrogen. Total RNA was extracted using the RNAprep Pure Cell/ Bacteria Kit (Tiangen Biotech, Beijing, China) following the manufacturer's protocol. RNA-seq libraries were prepared using the NEB-Next Ultra RNA Library Prep Kit, following the manufacturer's protocol, before sequencing on the Novaseq 6000 system at the Novogene Bioinformatics Institute (Beijing, China) to generate 150 bp paired-end reads. The NGS QC Toolkit v2.3.3 was used to filter low-quality reads from the raw data. For RPKM calculation, the M382 ORF index was constructed and the clean reads were mapped to the ORFs using Bowtie 2 v2.4.2[77]. To standardize the genetic expression, the number of reads mapped to each ORF was converted to RPKMs using Samtools v1.11[78]. A *P* value was <0.05 (two-sided Student's *t* test following a normality test) and an absolute value of $\log_2$ (fold change of RPKM) > 1 were used as the thresholds of significance. Differentially expressed genes were illustrated using volcano plots drawn in MS Excel and heatmaps drawn with Cluster 3.0 (hierarchical clustering with average linkage[79]) and Java TreeView[80].

### Cell membrane proteomics

The M382 biofilms used for membrane proteomics were cultured under the same conditions as those used for transcriptomics, as described above, except that four replicates were prepared. The cells were harvested at $5000 \times g$ for 5 min and subjected to membrane protein extraction using the bacterial membrane protein extraction kit HR0091 (Biomart, China). Protein concentrations were quantified using the BCA Protein Assay Kit (Bio-Rad, USA). The proteins were digested with trypsin, and the resulting peptides were desalted on C18 cartridges (Empore™ SPE Cartridges C18, Sigma). The desalted peptides were concentrated by vacuum centrifugation and dissolved in 40 μl of 0.1% (v/v) formic acid. Then, LC-MS/MS and protein identification was conducted in APTBIO (China). Briefly, LC-MS/MS analysis was conducted on a timsTOF Pro mass spectrometer (Bruker, Millerica, MA, USA), coupled with a Nanoelute (Bruker Daltonics). The peptides were loaded onto a reverse-phase trap column (Thermo Scientific Acclaim PepMap100, nanoViper C18) in buffer A (0.1% formic acid) and separated with buffer B (84% acetonitrile and 0.1% formic acid) at a flow rate of 300 nl/min. The ion mobility MS spectra were collected over a mass range of *m/z* 100–1700, and 10 PASEF MS/MS cycles were conducted with a target intensity of 1.5k and a threshold of 2,500. For protein identification and quantitation, the MS raw data were searched using the MaxQuant v1.5.3.17 software[81]. Protein discovery was set as a false discovery rate <0.01 and protein abundance was determined using the LFQ algorithm. Statistical analysis of protein abundance variations was based on two-sided Student's *t* test following a normality test, with significance thresholds of fold change >2 and *P* value <0.05.

## Measurement of membrane potential

The M382 biofilms used for PMF measurements were prepared under the same conditions as those used for transcriptomics and membrane proteomics. The bacterial cells were washed twice in phosphate-buffered saline (pH = 7.4) and resuspended to an $OD_{600}$ of 0.5. PMF was then measured using a membrane potential-sensitive probe DiSC3(5)[82]. DiSC3(5) and KCl were added to the cell solution to final concentrations of $1\,\mu m$ and $100\,mM$, respectively. The cells were incubated at room temperature for 30 min and fluorescence was recorded using a Biotek Cytation5 imaging reader (excitation wavelength = $622 \pm 10\,nm$ and emission wavelength = $670 \pm 10\,nm$).

## Statistics and reproducibility

In all the experiments, two-sided Students' $t$ tests were used to detect significant difference between the control and the experimental group. All experiments were performed for at least three times, each time with three or four biologically independent replicates.

## Reporting summary

Further information on research design is available in the Nature Portfolio Reporting Summary linked to this article.

## Data availability

The complete genome sequences (54 strains), metagenomic reads (6 samples), the metatranscriptomic reads (6 samples), and the transcriptomic datasets (6 samples) generated in this study have been deposited in the NCBI-SRA database under accession code PRJNA753157. The proteomics data (8 samples) have been deposited in the PRIDE database under accession code PXD040961. Source data are provided with this paper.

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

## Acknowledgements
This work is partially funded by grants from the following organizations/programs: (1) Ocean University of China (842041010); (2) Hong Kong Branch of Southern Marine Science and Engineering Guangdong Laboratory (SMSEGL20SC02); (3) Central Organization Department (862105020028); and (4) Shenzhen Basic Research General Program (JCYJ20210324122211031).

## Author contributions
W.Z. and Y.-X.L. designed the project and provided funding; W.D. performed all data analyses; S.W. and X.S. collected samples and isolated bacterial strains; P.Q., S.F., P.C., J.L., H.C., Y.S., Y.W., and M.W. performed experiments; Y.-Z.Z. and H.-H.F. provided guidelines and comments; W.Z. wrote the paper.

## Competing interests
The authors declare no competing interests.
