## [Peer Review File · Nature Communications]

Anaerobic thiosulfate oxidation by the Roseobacter group is prevalent in marine biofilmsReviewer #1 (Remarks to the Author):

Many congrats to the authors for this very interesting and highly relevant study on marine roseobacter bacteria, their role in biofilm formation and (an)aerobic thiosulfate oxidation. The work fits perfectly to the journal and is of high environmental relevance. Being not a bioinformatician, I will focus on the biological aspects in this review.

Wei Ding et al. started with the hypothesis that roseobacters represent important oxidizers of thiosulfate in marine biofilms. The beauty of this studies lies in the combination of metagenomic- and metatranscriptomic approaches with cultivation and physiological characterization of novel biofilm associated roseobacters. Thus, the authors isolated 54 novel biofilm-associated roseobacter strains, including one novel genus and determined their genome sequences. Compared to 95 NCBI genomes the phylogeny was determined based on 31 conserved single-copy genes. The so achieved novel linages call for valid description. The authors proofed the strain differences by ANI analysis and studied plasmids and sox gene clusters among the novel isolates obtained in this study. While most studies would have stopped at this point, the authors brought their findings into global context by comparing their data with 152 biofilm associated microbiota from databases and with their own metatranscriptomic data. Again, this is where most advanced studies would have stopped. However, the authors performed cultivation and thus were able to proof their computationally gained hypothesis in physiological experiments. Taken together, the authors nicely showed that roseobacters are the major players in thiosulfate oxidation in marine biofilms and that this oxidation is mostly done anaerobically with great significance to surface-associated bacteria.

Taken together the study could be published as it is. However, I have three suggestions the authors should feel free to address or to ignore:

1. Valid description of novel strains

The novel strains are of great interest for many different types of analysis thus as interspecies interactions. Thus, it would be highly desirable to validly describe them and to deposit the strains in a strain collection to make them available to the scientific community. I do understand that a species description, due to some standardized text elements, might ruin the nice flow of the manuscript. Thus, maybe this could be done in a follow up manuscript.

2. Rhamnose operon, microscopy and biofilm formation

Interestingly, the previously suggested rhamnose operon was missing in most strains, indicating that it plays not the suggested major role in roseobacter biofilm formation. This very interesting finding would be easily to test with the newly acquired strains as for example by crystal violet biofilm forming assay.

This further relates to the very basic TEM microscopy of the newly acquired strains. I feel this point would be much stronger if light microscopic images are included to demonstrate for example the typical rosette formation of some roseobacter species. Are the novel strains really behaving differently as suggested by TEM images? Could the biofilm formation be shown by SEM for example? Do the biofilms differ somehow compared to the previously observed once? Some roseobacter strains show a high level of morphological variation. Is this true for the new strains as well?

3. Least known group

I highly suggest rewording L271-272: An article from 2014 made the statement that the roseobacters are still on of the least known marine microbial groups. However, a search term 'roseobacter' in the NCBI database reveled 419 articles from 2014 till today. So, this statement is a bit outdated. However, roseobacters are important and need further in-depth investigation such as in this study.

Christian Jogler, FSU Jena, Germany

Reviewer #2 (Remarks to the Author):

The manuscript entitled: "Anaerobic thiosulfate oxidation by roseobacters is prevalent in marine biofilms" by Ding and co-authors explores the role played by the marine Roseobacter group in S₂O₃²⁻ oxidation. The oxidation of S species has been studied for long, with relevant roles of Roseobacters in seawater (Gonzalez et al 1999), soils (Teske et al 2000) and marine sediments (Lenk et al 2012). Surprisingly, thiosulfate oxidation in marine biofilms is poorly explored. Recently, the biofilm formation in the marine environment has become an interesting topic since plastic particles can be colonized by marine bacteria and form biofilms (plastisphere) with unexpected high diversity and functional potential. The study performed by Ding and co-authors is very complete, leaving little doubt that Roseobacter play an important role in anaerobic thiosulfate oxidation in marine biofilms. The authors provide evidences from isolates, metagenomes, metatranscriptomes, experiments, functional gene screenings, transcriptomics and membrane-proteomics. The paper is in general well written although more details in methodology need to be included.

As a very general comment on the paper, I missed comments on the uncertainties of the findings, results and techniques used. All data is selected to point towards thiosulfate oxidation by biofilm roseobacters, but many more data is obtained and neglected. For instance, the identity of the 446 isolated strains other than roseobacter and their capacity to oxidize thiosulfate, or the identity of the sox-harboring taxa in the analyzed metagenomes (and not only the abundance of the 54 isolated roseobacter), etc. Authors should comment on all data related to thiosulfate oxidation and frame results into a wider perspective.

In the introduction, the relevance of S₂O₃²⁻ oxidizers and the current gap of knowledge that this papers fills should be more stressed. Importantly, the environmental relevance of thiosulfate oxidation is not highlighted. Specificity of sox cluster genes for thiosulfate oxidation should be also included in the introduction.

The relevant role of Roseobacters in S metabolisms is well known, what does it make special for marine biofilms? I think this is an important question that is actually also answered in the paper but not well introduced in the introduction. Given the current mess in bacterial taxonomies, due in part to the tsunami informations from GTDB and MAGs' taxonomies, could the authors describe what Roseobacter group include? The novel strains found in this work is a very relevant finding for the field.

Results.

Line 79. What about the other isolates from biofilms?

Line 105. I was so surprised to observe in Figure 1 the almost absence of diversity of Phaeobacters. Is it for any specific reason?

Line 110. It is not clear whether the authors look for plasmid marker genes or biofilm formation marker genes.

Line 138. For curiosity, were the organic S degradation genes quantified (dmdA, dddL, dddD, etc)?

Line 156. What about the abundance and taxonomy of the sox genes in the metagenomes?

Line 168. Instead of using ;, make to sentences.

Line 173. Roseobacter have a copiotrophic lifestyle and their abundance in seawater is driven by high Chlorophylla concentrations and high nutrient concentrations. Were these conditions included in the seawater metagenomes? This could be a reason of the lower abundance as free-living bacteria?

Line 180. Were metatranscriptomes duplicated or triplicated? The location and time of sampling is not described here no in M&M.

Line 204. A "in a niche-specific way" refers to biofilm habitat?

Line 207. It should specify experiments with some of the isolated Roseobacters.

Line 256. How many remained unidentified?

Discussion. The fact that Roseobacter is more abundant under bloom conditions that were not the conditions selected for the seawater metagenomes could be included.

Line 312. Please, reference better the 80% result based on data provided.

Conclusions. I think the conclusions of the manuscript should be more adjusted to the actual results of the paper. The work is mostly based on isolated Roseobacters which most probably differ from the wild species. There is no analyses on biofilm MAGs harboring sox clusters, for instance, which would provide a closer picture of wild bacteria.

Methods.

Line 346. Was the isolation performed under anaerobic conditions? Under artificial sunlight radiation or PAR radiation (relevant in this group of marine bacteria harboring proteorhodopsins)? This is important information since the authors aimed to show the anaerobic thiosulfate oxidation.

Line 347. Was DNA extracted or cells submitted to any membrane compromise process?

Line 349. What was the length of Sanger reads? How 16S amplicons were annotated?

Line 357. Which type of Illumina was used? What was the length of the reads? Was it paired sequencing? Libraries sizes?

Line 362. What was the Illumina correction on PacBio sequencing exactly?

Line 377. Here the information about the software to build the phylogenetic tree should be included.

Line 380-389. It is not clear to me what and where the DNA is extracted from. Which Illumina was used? I do not understand the normalization of metaG data to 1e6 reads and trimming to 101 bp. The genomes used for mapping were the 54 isolated roseobacter? It should also be included in the caption of Figure 2.

Line 393. How was RNA preserved without any RNA later or similar? Which strains were selected for the biotin-labeled oligos selection? What was the procedure (kits, reagents, etc) to retain mRNA and conversion to cDNA? Which genomes were used in Bowtie2, cultures or MAGs? Marine? Regarding mutants, I think the work was good performed and described.

Cell membrane proteomics. Which % of proteins represented membrane-proteins? What was the loss of the process?

Cited References:

González, J. M., Kiene, R. P., & Moran, M. A. (1999). Transformation of sulfur compounds by an abundant lineage of marine bacteria in the α -subclass of the class Proteobacteria. *Applied and environmental microbiology*, 65(9), 3810-3819.

Teske, Andreas, et al. "Diversity of thiosulfate-oxidizing bacteria from marine sediments and hydrothermal vents." *Applied and Environmental Microbiology* 66.8 (2000): 3125-3133.

Lenk, Sabine, et al. "Roseobacter clade bacteria are abundant in coastal sediments and encode a novel combination of sulfur oxidation genes." *The ISME journal* 6.12 (2012): 2178-2187.

Reviewer #3 (Remarks to the Author):

The authors performed a multi-level study of thiosulfate oxidation in marine biofilms. They first isolated 54 roseobacter group strains and show that most of them contained the sox gene cluster for thiosulfate oxidation in their genomes.

Then, they looked for those genomes in 146 marine biofilm metagenomes (plus six additional ones obtained in this study), and compared their prevalence to that found in metagenomes of free-living microbes (155 Tara Oceans surface water samples, and 32 hydrothermal vent fluid samples). The 54 genomes of isolates accounted for 1% of surface ocean biofilm metagenomes and 0.08% of surface ocean free living metagenomes (on average). In hydrothermal vents, they accounted for 0.48% in biofilm compared to 0.07% in fluid (on average).

Then, they did metatranscriptome sequencing for 4 coastal biofilm communities and found that among energy metabolism related genes, sox genes ranked among the top 50%.

Rhodobacteraceae (not Roseobacters) accounted for 66-87% of soxX and 58-83% of soxA. They

extracted two of the seven sox genes (soxA and soxX) and show that 19% and 17% belong to the genus *Sulfitobacter*, and high percentages were also found belonging to *Roseobacter*, *Leisingera*, and *Phaeobacter*. The genomes of the 54 isolates recruited about 4% of transcripts from the coastal biofilm libraries.

The conclusion from this part of the study is that sox genes are present in the majority of their biofilm isolates, these isolates are much more abundant in marine biofilm metagenomes than in free-living surface water or hydrothermal vent metagenomes, and active transcription of sox genes was dominated by Rhodobacteraceae in four coastal biofilms.

Next, they looked for evidence of anaerobic thiosulfate oxidation. They grew six biofilm isolates from representative genera in MB and show that they could grow aerobically and anaerobically.

Then they selected 4 strains that contained the sox genes and investigated growth in artificial seawater medium with 10 mM thiosulfate. They show that these four strains grew both aerobically and anaerobically using thiosulfate as an energy source.

Finally they focused on strain M382. They constructed mutants for soxA and soxX and grew wildtype and mutants on artificial seawater medium both aerobically and anaerobically. The wild-type grew slightly better under aerobic conditions, and sulfate accordingly accumulated more under aerobic conditions. The two mutant strains grew even slightly better than the wild-type, but did not produce sulfate as expected. Therefore, growth cannot have been the result of thiosulfate oxidation, so the author's conclusion is doubtful.

Finally, they performed transcriptomics of strain M383 in biofilms on complex media with or without thiosulfate. The presence of thiosulfate resulted in increased expression of the sox gene cluster and genes related anaerobic respiration. Using proteomics of "thiosulfate treated cells" they show increased abundance of 304 proteins, including the seven SOX proteins. The authors conclude that these proteins were localized in the cell membrane during thiosulfate oxidation – does that imply that they were located in the cytoplasm during growth on other energy sources? They also found some other membrane-localized proteins for anaerobic respiration to be more abundant in "thiosulfate treated cells".

Major comments: The study seeks to demonstrate the importance of roseobacter group taxa for anaerobic thiosulfate oxidation in marine biofilms, compared to free-living communities. They approach this question by analyzing composition and activity of complete communities (metagenomes and metatranscriptomes) and studying the genome, growth behavior, gene expression and proteome of isolates obtained from coastal biofilms in China, focusing on 54, 6, 4 or 1 isolate, depending on the applied method. The two approaches are linked by attempts to detect both the genomes and transcriptomes of the isolates in marine metagenomes and metatranscriptomes.

The study is a textbook approach in microbial ecology, since it covers both culturable and not-yet-cultured microbes, genome and transcriptome, and even membrane proteome, and since it uses next generation sequencing with very high sequencing depth and state of the art analyses.

However, this approach comes with certain problems:

- (1) The different analyses are not consistently performed on the same set of samples. Finally the results from a single strain are generalized back to the biofilm communities in the ocean.
- (2) The study ignores some of the state of the art, e.g. regarding thiosulfate oxidation, abundance of sox genes in Rhodobacteraceae, taxonomy of roseobacter group.
- (3) The conclusions from the proteomics experiment are wild in my opinion.

Detailed comments

Ad (1) Consistency

- Why was the abundance and expression of sox genes and genes for anaerobic metabolism not analysed in ocean metagenomes and metatranscriptomes?
- Please clarify why only 4 of the 6 coastal biofilm communities were chosen for metatranscriptome analysis, and how they were selected?
- Please clarify why strain M382 was chosen for cultivation, gene knock-out, and proteomics?
- Was the transcriptomics experiment for strain M382 performed under aerobic or anaerobic conditions, planktonically or as a biofilm? Would you detect difference in gene expression between planktonically and biofilm grown cells?

Ad (3) State of the art

- A quick browse of the recent literature shows that various pathways for thiosulfate oxidation are known in marine systems.

- Sulfitobacter, Rhodobacter and Erythrobacter are long known to perform aerobic and anaerobic sulfur oxidation, particularly in sediments, which are regarded as biofilms I believe.
- Is there evidence that thiosulfate is an abundant metabolite in marine waters/sediments?
- Has it been shown before that addition of thiosulfate induces expression of the sox gene cluster?

Ad (3) Proteomics conclusions

- Fueling biofilm development: There is no general biofilm concept, the mechanisms are different depending on ecosystem and strain, as I am sure you know. The reference that you cite for the energy demand for biofilm formation may not be relevant for marine biofilms. Likewise, many references can be found demonstrating that biofilms represent an attempt to survive harsh conditions, lack of nutrients etc. which means they reflect a condition where energy is scarce, and less is needed under biofilm conditions compared to planktonic life.
- Remodeling of the cell membrane: I think this is a mis-interpretation of the data. An enzyme can be localized either in the cytoplasm or in the cell membrane, but it is not recruited from the cytoplasm to the membrane if its substrate is provided.
- Thiosulfate treated cells: In what respect were the conditions different to those used for cultivation in ocean water with or without thiosulfate? In which respect were they different to the conditions used for the transcriptome analysis? Were the cells grown as biofilms or planktonically?

Minor comments:

- The taxonomy of roseobacters has been revised based on 106 completed genomes [1]. Please check if your phylogeny contains representatives from the main clusters described there.
- "Roseobacter" is not a valid taxonomic term. Since roseobacters are not monophyletic, as previously assumed, the term "roseobacter group" should be used for marine Rhodobacteraceae.
- Importance of plasmids: While you have detected plasmids with biofilm-relevant genes, you did not analyse if these genes are less abundant in genomes. So the conclusion that plasmids harbor the – biofilm related – diversity of roseobacters is not valid.
- Which carbon source was used for the cultivation in ocean water?
- Was M382 cultivated planktonically or as a biofilm? Was the growth effect of thiosulfate different under those two conditions?
- How do you interpret the fact that the deletion mutants for soxX and soxA grew just as well, both aerobically and anaerobically, although they did not reduce thiosulfate?

1. Simon M, C Scheuner, JP Meier-Kolthoff, T Brinkhoff, I Wagner-Döbler, M Ulbrich, H-P Klenk, D Schomburg, J Petersen, M Göker. Phylogenomics of Rhodobacteraceae reveals evolutionary adaptation to marine and non-marine habitats. ISME J 2017; 11: 1483–1499.

Reviewer #1 (Remarks to the Author):

Many congrats to the authors for this very interesting and highly relevant study on marine roseobacter bacteria, their role in biofilm formation and (an)aerobic thiosulfate oxidation. The work fits perfectly to the journal and is of high environmental relevance. Being not a bioinformatician, I will focus on the biological aspects in this review.

Wei Ding et al. started with the hypothesis that roseobacters represent important oxidizers of thiosulfate in marine biofilms. The beauty of this studies lies in the combination of metagenomic- and metatranscriptomic approaches with cultivation and physiological characterization of novel biofilm associated roseobacters. Thus, the authors isolated 54 novel biofilm-associated roseobacter strains, including one novel genus and determined their genome sequences. Compared to 95 NCBI genomes the phylogeny was determined based on 31 conserved single-copy genes. The so achieved novel lineages call for valid description. The authors proofed the strain differences by ANI analysis and studied plasmids and sox gene clusters among the novel isolates obtained in this study. While most studies would have stopped at this point, the authors brought their findings into global context by comparing their data with 152 biofilm associated microbiota from databases and with their own metatranscriptomic data. Again, this is where most advanced studies would have stopped. However, the authors performed cultivation and thus were able to proof their computationally gained hypothesis in physiological experiments. Taken together, the authors nicely showed that roseobacters are the major players in thiosulfate oxidation in marine biofilms and that this oxidation is mostly done anaerobically with great significance to surface-associated bacteria. Taken together the study could be published as it is. However, I have three suggestions the authors should feel free to address or to ignore:

1. Valid description of novel strains

The novel strains are of great interest for many different types of analysis thus as interspecies interactions. Thus, it would be highly desirable to validly describe them and to deposit the strains in a strain collection to make them available to the scientific community. I do understand that a species description, due to some standardized text elements, might ruin the nice flow of the manuscript. Thus, maybe this could be done in a follow up manuscript.

Reply: The authors appreciated for your positive comments. We are doing the nomenclature for M382, including more details of its phenotype and biochemical characteristics. We will submit the manuscript to *International Journal of Systematic and Evolutionary Microbiology*.

2. Rhamnose operon, microscopy and biofilm formation

Interestingly, the previously suggested rhamnose operon was missing in most strains, indicating that it plays not the suggested major role in roseobacter biofilm formation. This very interesting finding would be easily to test with the newly acquired strains as for example by crystal violet biofilm forming assay.

This further relates to the very basic TEM microscopy of the newly acquired strains. I feel this point would be much stronger if light microscopic images are included to demonstrate for example the typical rosette formation of some roseobacter species. Are the novel strains really behaving differently as suggested by TEM images? Could the biofilm formation be shown by SEM for example? Do the biofilms differ somehow compared to the previously observed once? Some roseobacter strains show a high level of morphological variation. Is this true for the new strains as well?

Reply: Good suggestion! We have performed crystal violet staining on the biofilm formed by M382, and found strong biofilm formation ability on the bottom of six-well plates (**Figure for review 1**). We have also performed SEM observation on M382 and found certain rosette-like cells (**Figure for review 2**). These results suggested the ability of biofilm formation without rhamnase operon, and that the marine biofilm-derived *Roseobacter* strains may have similar behaviors as previous-known strains in terms of biofilm and rosette formation. However, because we don't have biofilm-forming *Roseobacter* strains reported by other groups, we could not perform comparative experiments to study the biofilm morphological features of these new *Roseobacter* strains.

Figure for review 1 Crystal violet staining on the biofilm formed by M382. Six-well plates without bacteria were used as negative controls. Six biological replicates were performed.

Figure for review 2 SEM observation on the M382 strains in a biofilm. Rosette-like cells are highlighted by circles.

3. Least known group

I highly suggest rewording L271-272: An article from 2014 made the statement that the roseobacters are still on of the least known marine microbial groups. However, a search term ‘roseobacter’ in the NCBI database reveled 419 articles from 2014 till today. So, this statement is a bit outdated. However, roseobacters are important and need further in-depth investigation such as in this study.

Reply: Revised according to the comments. Although most of the known members of this clade are culturable, *Roseobacter* strains are important and need further in-depth investigation due to the diversity of their habitats and metabolic activities.

Christian Jogler, FSU Jena, Germany

Reply: Thanks again for your support.

Weipeng Zhang

Reviewer #2 (Remarks to the Author):

The manuscript entitled: “Anaerobic thiosulfate oxidation by roseobacters is prevalent in marine biofilms” by Ding and co-authors explores the role played by the marine Roseobacter group in S₂O₃²⁻ oxidation. The oxidation of S species has been studied for long, with relevant roles of Roseobacters in seawater (Gonzalez et al 1999), soils (Teske et al 2000) and marine sediments (Lenk et al 2012). Surprisingly, thiosulfate oxidation in marine biofilms is poorly explored. Recently, the biofilm formation in the marine environment has become an interesting topic since plastic particles can be colonized by marine bacteria and form biofilms (plastisphere) with unexpected high diversity and functional potential. The study performed by Ding and co-authors is very complete, leaving little doubt that Roseobacter play an important role in anaerobic thiosulfate oxidation in marine biofilms. The authors provide evidences from isolates, metagenomes, metatranscriptomes, experiments, functional gene screenings, transcriptomics and membrane-proteomics. The paper is in general well written although more details in methodology need to be included.

As a very general comment on the paper, I missed comments on the uncertainties of the findings, results and techniques used. All data is selected to point towards thiosulfate oxidation by biofilm roseobacters, but many more data is obtained and neglected. For instance, the identity of the 446 isolated strains other than roseobacter and their capacity to oxidize thiosulfate, or the identity of the sox-harboring taxa in the analyzed metagenomes (and not only the abundance of the 54 isolated roseobacter), etc. Authors should comment on all data related to thiosulfate oxidation and frame results into a wider perspective.

Reply: Thanks a lot for your comments. In the revised manuscript, we have made substantial corrections by adding more analyses, new datasets and additional experiments to support the conclusion. We have also added many details into the method part.

We have analyzed the *sox* orthologs in biofilm metagenomes and found that a large percentage of the *sox* genes were affiliated to genera of the *Roseobacter* group, such as *Sulfitobacter*, *Ruegeria*, *Roseovarius*, *Phaeobacter*, *Roseibium*, and *Octadecabacter* (**Extended Data Fig. 12** in the revised manuscript and also shown as the following), suggesting that *Roseobacter* strains are indeed the major bacterial group contributing to thiosulfate oxidation in marine biofilms.

We have added two more metatranscriptomes to show the high expression of *Roseobacter*-affiliated *sox* genes. At the genus level, up to 21.05% of the *soxA* and up to 20.00% of the *soxX* genes (**Extended Data Fig. 15** in the revised manuscript and also shown as the following) were associated with *Sulfitobacter*, and these two genes also accounted for relatively high percentages in other genera of the *Roseobacter* group, such as *Roseobacter*, *Roseovarius*, and *Roseibium*. These results supported the notion that *Roseobacter* strains are the major contributors to thiosulfate oxidation in marine biofilms.

In addition, we have sequenced and annotated 304 genomes. As a result, we found 82 genomes with the *sox* gene clusters, 74 (90.2%) of which belonged to the *Roseobacter* group, including *Phaeobacter*, *Sulfitobacter*, *Leisingera*, *Aliiroseovarius*, *Limimaricola*, *Jannaschia*, *Pelagimonas*, *Yoonia*, and one unclassified strains (**Figure for review 2**). Because we have not finished sequencing of all the isolated strains, genomes of the other isolates (n > 500) will be published in a following study (hopefully a big publication). Nevertheless, all the results of genomic, metagenomic, and metatranscriptomic analyses

indicated *Roseobacter* strains as the major thiosulfate oxidizers in marine biofilms.

We have also added a statement of the limitation of this study. Limitations exist for the current study. For example, here we have focused on culturable *Roseobacter* strains in marine biofilms while metagenome-assembled genomes were not used. This is largely due to the high microbial diversity in marine biofilms, which hinders the recovery of high-quality genomes from biofilm metagenomes.

Extended Data Fig. 12 Taxonomic affiliation of the *soxX* and *soxA* genes in assembled biofilm metagenomes at genus level. The six biofilms collected in the present study were used for analyses. The affiliations were determined by searching against the *kaiju_db* database using Kaiju. **a**, *soxX*; **b**, *soxA*.

Extended Data Fig. 15 Taxonomic affiliation of the *soxX* and *soxA* genes in assembled biofilm metatranscriptomes at genus level. The six biofilms collected in the present study were used for analyses. The affiliations were determined by searching against the *kaiju_db* database using Kaiju. **a**, *soxX*; **b**, *soxA*.

Figure for review 1 Genus-level classification of all the genome-sequenced strains (n = 304).

Figure for review 2 Genus-level classification of all the genomes (n = 82) with the *sox* gene clusters. Of these genomes, 74 belonged to the Roseobacter group, including *Phaeobacter*, *Sulfitobacter*, *Leisingera*, *Aliiroseovarius*, *Limimanicola*, *Jannaschia*, *Pelagimonas*, *Yoonia*, and one unclassified strain.

In the introduction, the relevance of S₂O₃²⁻ oxidizers and the current gap of knowledge that this papers fills should be more stressed. Importantly, the environmental relevance of thiosulfate oxidation is not highlighted. Specificity of *sox* cluster genes for thiosulfate oxidation should be also included in the introduction.

Reply: We have added more sentences to describe the specificity of *sox* in thiosulfate oxidation. The influence of environmental condition on thiosulfate oxidation has been added. The major gap we want to fill is also highlighted.

First found in the lithoautotrophic bacteria *Paracoccus pantotrophus*, the sulfur oxidizing enzyme (SOX) system, a typical periplasmic multi-enzyme system, is the most well-known mediators of thiosulfate oxidation. Regarding electron acceptor utilization and niche distribution, aerobic thiosulfate oxidizers have been isolated from the deep chlorophyll maximum, coastal seawater, and coastal marine

sediments, whereas anaerobic thiosulfate oxidizers have been recently reported in anoxic marine basins, oceanic oxygen minimum zones, and hydrothermal vents. In addition, environmental conditions affect the major products of thiosulfate production. For example, in coastal marine sediments, thiosulfate is oxidized to varying proportions of tetrathionate and sulfate, as well as elemental sulfur, depending on the sulfidic and oxygenic conditions.

Specific questions are which bacteria are the greatest contributors to thiosulfate oxidation in marine biofilms and how they conduct this process.

The relevant role of Roseobacters in S metabolisms is well known, what does it make special for marine biofilms? I think this is an important question that is actually also answered in the paper but not well introduced in the introduction. Given the current mess in bacterial taxonomies, due in part to the tsunami informations from GTDB and MAGs' taxonomies, could the authors describe what Roseobacter group include? The novel strains found in this work is a very relevant finding for the field.

Reply: Good suggestion! The special physical structure of biofilms is probably the major reason for the existence of novel sulfur metabolism processes. Given thiosulfate oxidation is affected by the availability of electron acceptor, the oxygen gradient in biofilms is likely to breed novel thiosulfate-oxidizing bacteria. However, there is no general conclusion about the major thiosulfate-oxidizing taxa in global ocean, as we have discussed in the discussion part, and little experimental evidence exists for thiosulfate oxidation by *Roseobacter* strains.

In terms of the *Roseobacter* group, we have added more statement in the introduction. Bacteria of the *Roseobacter* group (or the Roseobacteraceae family), sharing > 89% identity in the 16S rRNA gene, are heterotrophs found worldwide in marine ecosystems. The so far discovered *Roseobacter* group comprises 327 species and 128 genera, represented by *Ruegeria* (e.g, *Ruegeria pomeroyi* DSS-3, a model strain), *Phaeobacter* (well-known strains for the production of tropodithietic acid), and *Sulfitobacter* (widely-distributed strains in the global ocean).

Results.

Line 79. What about the other isolates from biofilms?

Reply: At this moment, we have sequenced the genomes of 304 biofilm-derived strains. All the genomes got 98%-100% completeness. The following figure (**Figure for review 1** shown above) shows the family-level profile of all the genome-sequenced strains. To address the reviewer's question, we annotated these genomes by searching against the KEGG database (2022 version). As a result, we found 82 genomes with the *sox* gene clusters, 74 (90.2%) of which belonged to the *Roseobacter* group (**Figure for review 2** shown above). These results are largely consistent with the metagenomic and metatranscriptomic analyses. Since the focus of the current study is *Roseobacter*, and we have not finished genome sequencing of all the genomes, we only documented the 54 *Roseobacter* genomes in the present manuscript. Genomes of the other isolates will be published in a following study.

Line 105. I was so surprised to observe in Figure 1 the almost absence of diversity of *Phaeobacters*. Is it for any specific reason?

Reply: Good suggestion! The reasons might be the strong biofilm formation ability and the production of tropodithietic acid (TDA). *Phaeobacter inhibens* strains are effective biofilm formers and colonizers of marine surfaces and have the ability to outcompete other microbes. Therefore, the presence of other *Phaeobacters* in the biofilm communities is likely to be prevented. However, more efforts are needed to illustrate this notion.

References:

Gram et al. *Phaeobacter inhibens* from the Roseobacter clade has an environmental niche as a surface colonizer in harbors Syst Appl Microbiol. 2015. 38:483-93

Zhao et al. Contributions of tropodithietic acid and biofilm formation to the probiotic activity of *Phaeobacter inhibens*. BMC Microbiol. 2016. 16:1.

Line 110. It is not clear whether the authors look for plasmid marker genes or biofilm formation marker genes.

Reply: Both. Based on previous studies (May and Okabe, 2008; Michael et al. 2016), genes on plasmids are often important for biofilm formation. For example, the conjugative machinery of plasmid F stimulates *E. coli* to synthesize colanic acid and curli proteins, which play a role in biofilm maturation. For *Roseobacter* strains, rhamnase operon in plasmids play roles in biofilm formation.

References:

May and Okabe. *Escherichia coli* harboring a natural *incF* conjugative *f* plasmid develops complex mature biofilms by stimulating synthesis of colanic acid and curli. J Bacteriol. 2008. 190:7479–7490.

Michael et al. Biofilm plasmids with a rhamnase operon are widely distributed determinants of the ‘swim-or-stick’ lifestyle in roseobacters. ISME J. 2016. 10:2498-2513.

Line 138. For curiosity, were the organic S degradation genes quantified (*dmdA*, *dddL*, *dddD*, etc)?

Reply: Good question! Most of these biofilm *Roseobacter* strains possess DMSP metabolism-related genes. Because DMSP is not the focus of the present study, we didn’t present the relevant results.

Line 156. What about the abundance and taxonomy of the *sox* genes in the metagenomes?

Reply: We have added this result in the revised manuscript. Please see **Extended Data Fig. 12** displayed above.

Line 168. Instead of using ;, make to sentences.

Reply: Revised.

Line 173. *Roseobacter* have a copiotrophic lifestyle and their abundance in seawater is driven by high

Chlorophylla concentrations and high nutrient concentrations. Were these conditions included in the seawater metagenomes? This could be a reason of the lower abundance as free-living bacteria?

Reply: Good question! In a relevant study which will be submitted to another journal in the coming weeks, we explored the effect of carbon source concentration on growth of M382 (please see **Figure for review 3**). Consistent with the prediction by the reviewer, we demonstrated the copiotrophic lifestyle for M382, but not for its free-living relative, DSS-3. Therefore, accumulation of high organic materials in biofilms may well explain the prevalence of these strains in biofilms rather than in seawater.

Figure for review 3 Growth features of M382 and DSS-3 at different concentrations of carbon source. **a,b** Growth kinetics of M382 and DSS-3 when cultured planktonically in 2216E and 1/10 2216E media. **c,d** Scanning electron micrographs of biofilms formed by M382 in 2216E and 1/10 2216E media. **e** Maximum cell density of M382 biofilms grown in 2216E and 1/10 2216E media (**P-value < 0.01 in two-tailed Student's t-test). The error bar represents the standard deviation of three biological replicates.

Line 180. Were metatranscriptomes duplicated or triplicated? The location and time of sampling is not described here no in M&M.

Reply: Due to the high sequencing cost, we only included one replicate. However, because the six

metatranscriptomes were collected from different seasons, we believe that the high expression of *Roseobacter*-affiliated *sox* genes can support our conclusion. Details (e.g., sampling location and time) have been added to the method part of the revised manuscript and are also shown below.

In addition to metagenomes downloaded from previous studies, six biofilms were newly collected in the present study, followed by metagenomic and metatranscriptomic sequencing. These biofilms were scraped from subtidal stone surfaces in the coastal area (36.05, 120.43) of Qingdao, China, and in total six time points (Sep 2020, Nov 2020, Jan 2021, Mar 2021, May 2021, and Jul 2021) were selected for sampling.

Line 204. A “in a niche-specific way” refers to biofilm habitat?

Reply: To avoid misleading and overstatement, we removed this sentence.

Line 207. It should specify experiments with some of the isolated *Roseobacters*.

Reply: Revised. Before conducting thiosulfate oxidation experiments for representative strains of the biofilm *Roseobacter* strains, we explored their general physiological characteristics. Besides, in the revised manuscript, we have added more results about the phenotype and the thiosulfate oxidation of these strains. In total, nine strains representing the nine different genera were included.

Line 256. How many remained unidentified?

Reply: Revised by adding more details. In total, 68 KEGG genes (genes that could be annotated by KEGG) were significantly (fold-change > 2 and P-value < 0.05 by Student’s t-test) up-regulated by thiosulfate, 58 KEGG genes were down-regulated, while 4,607 KEGG genes remained unchanged (see **Extended Data Fig. 21** in the revised manuscript and also displayed below).

Extended Data Fig. 21 Overview of gene transcriptome profiles in M382 biofilms cultured with or without thiosulfate. Statistical analysis based on RPKM values was performed using Student's t-test with a threshold of fold change > 2 and P-value < 0.05. The up-regulated, down-regulated, and unchanged genes were colored in red, blue, and grey, respectively.

Discussion.

The fact that *Roseobacter* is more abundant under bloom conditions that were not the conditions selected for the seawater metagenomes could be included.

Reply: The biofilm and seawater metagenomes were collected across different seasons. We found that the abundance of these biofilm-derived *Roseobacter* strains are always higher in the biofilms than in the seawater, suggesting that their enrichment in the biofilms is not majorly affected by bloom conditions. We have added this statement to the discussion part.

Line 312. Please, reference better the 80% result based on data provided.

Reply: This result comes from taxonomic analysis of the *sox* genes in biofilm metagenomes and metatranscriptomes. We have revised this statement to avoid misunderstanding.

Conclusions.

I think the conclusions of the manuscript should be more adjusted to the actual results of the paper. The work is mostly based on isolated *Roseobacters* which most probably differ from the wild species. There is no analyses on biofilm MAGs harboring sox clusters, for instance, which would provide a closer picture of wild bacteria.

Reply: Good suggestion! We analyzed the MAGs reported in Zhang et al. (Nature Communications, 2019). These MAGs were binned from the global biofilm metagenomes and their taxonomy is shown as the following (**Figure for review 4**). We found that of the 24 *Roseobacter* MAGs, 16 belonged to genera (*Sufitobacter*, *Yoonia*, *Alliroseovarius*, *Phaeobacter*, and *Jannaschia*) that have been isolated in the current study. However, the quality of these MAGs is not high (~50% completeness and without plasmids), which hinders an in-depth analyses. Actually, we have spent a lot of efforts to get more information of these ‘wild species’ by using PacBio and Nanopore sequencing, but it turned out genome binning could not be well conducted on marine biofilms, probably due to the high microbial richness. Moreover, without cultured strains, no experiments could be performed to give solid evidence. In addition, a number of studies (Kent et al., 2018; Sharpe et al., 2020) have used isolated strains as models to study the ecological and physiological functions of the *Roseobacter* group, suggesting that the isolated *Roseobacter* strains can keep their ‘wild features’ to a large extent. Considering all these reasons, here we focused on isolated *Roseobacter* strains to explore their functions. We have added a statement of the limitation of this study (shown in the discussion part of the manuscript and as the following).

Limitations exist for the current study. For example, here we have focused on culturable *Roseobacter* strains in marine biofilms while metagenome-assembled genomes were not used. This is largely due to the high microbial diversity in marine biofilms, which hinders the recovery of high-quality genomes from biofilm metagenomes.

References:

Sharpe GC, Gifford SM, Septer AN. A model *Roseobacter*, *Ruegeria pomeroyi* DSS-3, employs a diffusible killing mechanism to eliminate competitors. *mSystems*. 2020. 5:e00443-20

Kent AG, Garcia CA, Martiny AC. Increased biofilm formation due to high-temperature adaptation in marine *Roseobacter*. *Nat Microbiol*. 2018. 3:989-995.

Zhang, W., et al. Marine biofilms constitute a bank of hidden microbial diversity and functional potential. *Nat. Commun*. 2019. 10:517.

Figure for review 4 Taxonomic classification of *Roseobacter* MAGs derived from global biofilm metagenomes. The classification was conducted on the genus level.

Methods.

Line 346. Was the isolation performed under anaerobic conditions? Under artificial sunlight radiation or PAR radiation (relevant in this group of marine bacteria harboring proteorhodopsins)? This is important information since the authors aimed to show the anaerobic thiosulfate oxidation.

Reply: Good question! We have added more details into the revised manuscript. Microbial incubations were conducted under 25 °C and aerobic condition with artificial sunlight radiation. Based on our experience, most of the biofilm-sourced microbes are facultative, meaning that they can grow at both aerobic and anaerobic conditions although all the strains were isolated under aerobic condition.

Line 347. Was DNA extracted or cells submitted to any membrane compromise process?

Reply: We have added more details into the revised manuscript. Conspicuous colony types were isolated and DNA was released from cells by heating at 100 °C for 5 minutes.

Line 349. What was the length of Sanger reads? How 16S amplicons were annotated?

Reply: We have added more details into the revised manuscript. The read length generated in one reaction is about 800 bp. So, for each strain, both forward and reverse reactions were performed to get the full-length 16S rRNA gene sequence (~1,500 bp).

Line 357. Which type of Illumina was used? What was the length of the reads? Was it paired sequencing? Libraries sizes?

Reply: We have added more details into the revised manuscript. The libraries (insert length = 350 bp) were sequenced on the Novaseq 6000 system in Novogene (Beijing, China) to generate 60 Gb of data (paired-end reads with a read length of 150 bp) for each biofilm sample.

Line 362. What was the Illumina correction on PacBio sequencing exactly?

Reply: We have added more details into the revised manuscript. Preliminary assembly and correction were performed with SMRT Link v5.0.1 (CCS = 3, minimum accuracy > 0.99), and then corrected by the Illumina data using Minimap2 (Minimap2: pairwise alignment for nucleotide sequences). In detail, the PacBio-derived contigs were mapped with the Illumina reads, and the locations without alignment was corrected according to the contigs assembled from Illumina reads.

Line 377. Here the information about the software to build the phylogenetic tree should be included.

Reply: We have added more details into the revised manuscript. The tree was built in MEGA and bootstrap values were calculated with 500 replicates.

Line 380-389. It is not clear to me what and where the DNA is extracted from. Which Illumina was used? I do not understand the normalization of metaG data to 1e6 reads and trimming to

Reply: We have added more details into the revised manuscript. In addition to metagenomes downloaded from previous studies, six biofilms were newly collected in the present study, followed by metagenomic and metatranscriptomic sequencing. These biofilms were scraped from subtidal stone surfaces in the coastal area (36.05, 120.43) of Qingdao, China, and in total six time points (Sep 2020, Nov 2020, Jan 2021, Mar 2021, May 2021, and Jul 2021) were selected for sampling. Metagenomic sequencing was performed on the NovaSeq 6000 system. Metatranscriptomes were also sequenced on the Novaseq 6000 system. Normalization is necessary, because some of the metagenomes downloaded from NCBI have different read length (e.g., 101 bp), and the total number of reads used for mapping can also affect the results.

101 bp. The genomes used for mapping were the 54 isolated roseobacter? It should also be included in the caption of Figure 2.

Reply: Yes. Revised. Figure 2. Global distribution of *Roseobacter* strains in biofilm-associated and free-living microbiota. The distribution pattern was drawn by mapping reads from 339 (152 biofilm and mat metagenomes versus 187 seawater and hydrothermal vent-fluid metagenomes) to chromosomes of the 54 biofilm-derived *Roseobacter* strains using BBMap (minimum alignment identity = 0.80). All the metagenomes were normalized to 1,000,000 reads with a read length of 101 bp.

Line 393. How was RNA preserved without any RNA later or similar? Which strains were selected for the biotin-labeled oligos selection? What was the procedure (kits, reagents, etc) to retain mRNA and conversion to cDNA? Which genomes were used in Bowtie2, cultures or MAGs? Marine?

Reply: Because the high density of RNA later prevents cell collection by centrifugation, the sequencing

company didn't recommend to use RNA later. Instead, we freeze the samples by adding liquid nitrogen and based on our experience, this is a good way to keep the integrity of RNA. Because universal oligos were used, rRNA and tRNA of all the bacterial and archaeal strains were selected and removed. The kit used for library construction was the NEBNext Ultra RNA Library Prep Kit (NEB, USA).

To determine the percentage of the *Roseobacter* strains, the metatranscriptomic reads were mapped to the 54 genomes using BMap (minimum alignment identity = 0.80) and the relative abundance was calculated by counting the numbers of mapped reads. To determine the expression level and taxonomic affiliation of a given gene in the metatranscriptomes, clean metatranscriptomic reads were assembled using MEGAHIT. ORFs were predicted from the assembled contigs using Prodigal in a Meta mode, and only close-ended ORFs were used for further analysis. The ORFs were annotated by DIAMOND BLASTp (E-value < 1e-7) searching against the KEGG database (2022 version). The clean metatranscriptomic reads were mapped to ORF sequences using Bowtie2 v2.4.2. The coverage of a given gene (e.g., *soxA*) was calculated using SAMtools to determine the gene expression profile, displayed in RPKM. Taxonomic affiliation at the genus levels was profiled using Kaiju, with the kaiju_db as the reference.

Regarding mutants, I think the work was good performed and described.

Reply: Thanks! We cannot agree with you more.

Cell membrane proteomics. Which % of proteins represented membrane-proteins? What was the loss of the process?

Reply: Very good question! Membrane proteins were extracted using a bacterial membrane protein extraction kit HR0091 (Biomart, China). In total we identified 3,050 proteins by using the membrane proteomics. However, due to the protein-protein interactions, it's difficult to determine which proteins are really distributed in the membrane through this proteomics. Nevertheless, a number of abundant proteins identified in the present study are previously known as membrane proteins, such as SOX proteins and general secretion pathway proteins, suggesting that the membrane proteomics results are reliable. In addition, our analysis is confined to these abundant proteins of interest, while ignoring low-abundance proteins that may not be real membrane proteins. The loss of this process would be certain low-abundance proteins that cannot be detected through MS.

Cited References:

González, J. M., Kiene, R. P., & Moran, M. A. (1999). Transformation of sulfur compounds by an abundant lineage of marine bacteria in the α -subclass of the class Proteobacteria. Applied and environmental microbiology, 65(9), 3810-3819.

Teske, Andreas, et al. "Diversity of thiosulfate-oxidizing bacteria from marine sediments and hydrothermal vents." Applied and Environmental Microbiology 66.8 (2000): 3125-3133.

Lenk, Sabine, et al. "Roseobacter clade bacteria are abundant in coastal sediments and encode a novel combination of sulfur oxidation genes." The ISME journal 6.12 (2012): 2178-2187.

Reply: These references are also closely related to our work and have been cited.

Thanks again for your comments.

Weipeng Zhang

Reviewer #3 (Remarks to the Author):

The authors performed a multi-level study of thiosulfate oxidation in marine biofilms. They first isolated 54 roseobacter group strains and show that most of them contained the sox gene cluster for thiosulfate oxidation in their genomes. Then, they looked for those genomes in 146 marine biofilm metagenomes (plus six additional ones obtained in this study), and compared their prevalence to that found in metagenomes of free-living microbes (155 Tara Oceans surface water samples, and 32 hydrothermal vent fluid samples). The 54 genomes of isolates accounted for 1% of surface ocean biofilm metagenomes and 0.08% of surface ocean free living metagenomes (on average). In hydrothermal vents, they accounted for 0.48% in biofilm compared to 0.07% in fluid (on average). Then, they did metatranscriptome sequencing for 4 coastal biofilm communities and found that among energy metabolism related genes, sox genes ranked among the top 50%. Rhodobacteraceae (not Roseobacters) accounted for 66-87% of soxX and 58-83% of soxA. They extracted two of the seven sox genes (soxA and soxX) and show that 19% and 17% belong to the genus Sulfitobacter, and high percentages were also found belonging to Roseobacter, Leisingera, and Phaeobacter. The genomes of the 54 isolates recruited about 4% of transcripts from the coastal biofilm libraries. The conclusion from this part of the study is that sox genes are present in the majority of their biofilm isolates, these isolates are much more abundant in marine biofilm metagenomes than in free-living surface water or hydrothermal vent metagenomes, and active transcription of sox genes was dominated by Rhodobacteraceae in four coastal biofilms. Next, they looked for evidence of anaerobic thiosulfate oxidation. They grew six biofilm isolates from representative genera in MB and show that they could grow aerobically and anaerobically. Then they selected 4 strains that contained the sox genes and investigated growth in artificial seawater medium with 10 mM thiosulfate. They show that these four strains grew both aerobically and anaerobically using thiosulfate as an energy source.

Reply: Thank you very much for your comments.

They focused on strain M382. The constructed mutants for soxA and soxX and grew wildtype and mutants on artificial seawater medium both aerobically and anaerobically. The wild-type grew slightly better under aerobic conditions, and sulfate accordingly accumulated more under aerobic conditions. The two mutant strains grew even slightly better than the wild-type, but did not produce sulfate as expected. Therefore, growth cannot have been the result of thiosulfate oxidation, so the author's conclusion is doubtful.

Reply: This is a very good question. In the revised manuscript, we examined the growth of wild-type M382, $\Delta soxX$, and $\Delta soxA$ in the biofilm-forming state. After cultured under aerobic or anaerobic condition in artificial seawater media with 10 mM thiosulfate, the wild-type strain displayed significantly higher cell density than the two mutants (shown in **Extended Data Fig. 20** in the revised manuscript and also displayed below), suggesting that thiosulfate oxidation may affect the bacterial growth when they are in biofilm state rather than grown planktonically.

Finally, they performed transcriptomics of strain M382 in biofilms on complex media with or without thiosulfate. The presence of thiosulfate resulted in increased expression of the sox gene cluster and genes related anaerobic respiration. Using proteomics of "thiosulfate treated cells" they show increased

abundance of 304 proteins, including the seven SOX proteins. The authors conclude that these proteins were localized in the cell membrane during thiosulfate oxidation – does that imply that they were located in the cytoplasm during growth on other energy sources? They also found some other membrane-localized proteins for anaerobic respiration to be more abundant in “thiosulfate treated cells”.

Reply: This is also a very good question. Our results suggested that the relative abundance of SOX proteins in the cell membrane are higher after adding thiosulfate, which could be attributed to their higher expression. However, we could not conclude that the cellular location of SOX proteins can be changed by thiosulfate. We have revised relevant statement in the manuscript to avoid misleading.

Major comments: The study seeks to demonstrate the importance of roseobacter group taxa for anaerobic thiosulfate oxidation in marine biofilms, compared to free-living communities. They approach this question by analyzing composition and activity of complete communities (metagenomes and metatranscriptomes) and studying the genome, growth behavior, gene expression and proteome of isolates obtained from coastal biofilms in China, focusing on 54, 6, 4 or 1 isolate, depending on the applied method. The two approaches are linked by attempts to detect both the genomes and transcriptomes of the isolates in marine metagenomes and metatranscriptomes.

Reply: We have made substantial corrections according to these questions. In the revised manuscript, we have added results of new experiments. We analyzed the complete genomes and examined the growth of 54 *Roseobacter* strains. SEM observation and thiosulfate oxidation experiment have been conducted on 9 strains representing the 9 different genera (shown in **Extended Data Figs. 17 and 19** and also displayed below). We further performed experiments including gene knock out, proteomics, transcriptomics, as well as biochemical experiments (e.g., PMF measurement) on 1 strain, which is probably represent a new genus under the *Roseobacter* group. It is indeed difficult to include all the experiments on all the strains in one manuscript. Our future studies will investigate features other strains, including sulfur oxidation. For example, in a new manuscript which will be submitted in the coming weeks, we analyzed the influence of temperature increase on gene expression in the biofilm of *Leisingera aquaemixtae* M597 (please see **Figure for review 1** displayed below). In addition, in another manuscript which will also be submitted in the coming weeks, we studied the influence of carbon source concentration on gene expression of M382, and the results revealed potential correlation between carbon source concentration and expression of the *sox* genes (please see **Figure for review 2** displayed below).

The study is a textbook approach in microbial ecology, since it covers both culturable and not-yet-cultured microbes, genome and transcriptome, and even membrane proteome, and since it uses next generating sequencing with very high sequencing depth and state of the art analyses.

Reply: The authors appreciate for your positive comments.

However, this approach comes with certain problems:

- (1) The different analyses are not consistently performed on the same set of samples. Finally the results from a single strain are generalized back to the biofilm communities in the ocean.
- (2) The study ignores some of the state of the art, e.g. regarding thiosulfate oxidation, abundance of *sox* genes in Rhodobacteraceae, taxonomy of roseobacter group.

(3) The conclusions from the proteomics experiment are wild in my opinion.

Reply: Thanks for your comments to improve the quality of this study. In the revised manuscript, we have added 2 additional biofilm metatranscriptomes. We also performed more analyses on *sox* genes and anaerobic respiration-related genes. The taxonomic affiliations of *soxA*, *soxX*, *napA*, and *nirK* genes in metagenomes and metatranscriptomes have been analyzed and added (shown in **Extended Data Figs. 12, 13, 15, 16** and also displayed below). The results suggested high abundance and expression of *Roseobacter*-associated *soxA*, *soxX*, and *nirK* genes in the biofilms, while the majority of *napA* genes are not affiliated to the *Roseobacter* group.

In the revised manuscript, we performed transcriptomics experiment for strain M382 living planktonically. As a result (shown below in **Figure for review 3**), we found that when grown planktonically, thiosulfate could not induce the expression of *sox* genes. This results suggests that the induction of *sox* genes by thiosulfate is in a biofilm-specific way.

In the revised manuscript, we also performed proton motive force (PMF) measurement experiment to show that thiosulfate is likely to be used as energy. Based on the increased abundance of proteins for anaerobic respiration and biofilm formation, we speculated the thiosulfate is used as an energy in the biofilm state. To test this speculation, we measured PMF in the M382 biofilms with or without thiosulfate. As a result, adding thiosulfate increased PMF production (shown in **Extended Data Fig. 25** and also displayed below). This result is consistent with the higher cell density of M382 biofilms when thiosulfate is present (shown in **Extended Data Fig. 20** and also displayed below). In addition, we have revised proteomics-related statement in the manuscript to avoid overstatement.

Extended Data Fig. 12 Taxonomic affiliation of the *soxX* and *soxA* genes in assembled biofilm metagenomes at genus level. The six biofilms collected in the present study were used for analyses. The affiliations were determined by searching against the *kaiju_db* database using Kaiju. **a**, *soxX*; **b**, *soxA*.

Extended Data Fig. 13 Taxonomic affiliation of the *napA* and *nirK* genes in assembled biofilm metagenomes at genus level. The six biofilms collected in the present study were used for analyses. The affiliations were determined by searching against the *kaiju_db* database using Kaiju. **a**, *napA*; **b**, *nirK*.

Extended Data Fig. 15 Taxonomic affiliation of the *soxX* and *soxA* genes in assembled biofilm metatranscriptomes at genus level. The six biofilms collected in the present study were used for analyses. The affiliations were determined by searching against the *kaiju_db* database using Kaiju. **a**, *soxX*; **b**, *soxA*.

Extended Data Fig. 16 Taxonomic affiliation of the *napA* and *nirK* genes in assembled biofilm metatranscriptomes at genus level. The six biofilms collected in the present study were used for analyses. The affiliations were determined by searching against the kaiju_db database using Kaiju. **a**, *napA*; **b**, *nirK*.

Extended Data Fig. 17 Transmission electron microscopy observation of selected biofilm *Roseobacter* strains. Nine strains from the nine distinct genera were observed at 10,000-50,000 times magnification. Scale bar = 500 nm.

Extended Data Fig. 19 Thiosulfate oxidation by representative biofilm *Roseobacter* strains. Sulfate production in nine strains from the nine distinct genera were detected. Strains were grown in minimum media with 10 mM thiosulfate for 72 hours under aerobic and anaerobic conditions.

Extended Data Fig. 20 Growth of wild-type M382 and its two *sox* gene mutants. The strains were grown in the biofilm state in minimum media with 10 mM thiosulfate, followed by measurement of cell density at 600 nm after cultivation for 72 hours. Both aerobic (a) and anaerobic (b) conditions were studied. Two-tailed Student's t-test was used to detect the significant difference (**P-value < 0.01; ***P-value < 0.001).

Extended Data Fig. 25 Proton motive forces (PMFs) in M382 biofilms grown with or without thiosulfate. The PMF was indicated by the fluorescence intensity of DiSC3(5). Two-tailed Student's t-test was used to examine significant differences (***)P-value < 0.001). Error bars represent standard deviation of three biological replicates.

Figure for review 1 Transcriptional analysis of free-living and biofilm-associated M597 cultivated at different temperatures. **A** The common and specific differentially expressed genes (DEGs) under different temperatures in the free-living and biofilm-associated cells. The up arrow indicates up-regulated genes in 25 °C whereas the down arrow indicates up-regulated genes in 31 °C. The cutoff for differential gene expression is P-value < 0.05 in Student’s t-test and fold-change of reads per kilobase per million mapped reads (RPKMs) > 1.5. **B** The overall distribution pattern of temperature-dependent DEGs in free-living and biofilm-associated M597 cells. ‘FC’ indicates fold change of RPKMs. **C** Principal component analysis reflecting the ordination of gene expression in different groups. **D** Heatmap of representative DEGs revealed by the comparison between 25 °C free-living versus 31 °C free-living cells and by the comparison between 25 °C biofilm-associated versus 31 °C biofilm-associated cells. RPKMs are indicated by a color gradient. The *sox* genes were differentially regulated by temperature increase in between biofilms and the free-living cells.

Figure for review 2 Comparative analysis of M382 biofilms grown at two carbon source concentrations. Significantly altered genes (Student's t-test, P-value < 0.05 and RPKM fold change > 2) belonging to central carbon, amino acid, and sulfur metabolism pathways are shown.

Figure for review 3 Transcriptomic analyses of the free-living M382 cells in response to thiosulfate. M382 strains were grown in triplicate cultures in marine broth 2216 with or without 10 mM thiosulfate. Tubes with the cultures were incubated at 25 °C and 100 rpm for 48 hours. The datasets have been uploaded to the SRA database.

Detailed comments

Ad (1) Consistency

- Why was the abundance and expression of *sox* genes and genes for anaerobic metabolism not analysed in ocean metagenomes and metatranscriptomes?

Reply: We have added analyses according to your comments. The taxonomic affiliations of *soxA*, *soxX*, *napA*, and *nirK* genes in metagenomes and metatranscriptomes have been analyzed and added (shown in **Extended Data Figs. 12, 13, 15, 16**). The results revealed high abundance and expression of *Roseobacter*-associated *soxA*, *soxX*, and *nirK* genes in the biofilms, while the majority of *napA* genes was not affiliated to the *Roseobacter* group. Thus, nitrite rather than nitrate might be used as electron acceptor in thiosulfate oxidation by *Roseobacter* strains. To further address this question, we will perform *in situ* experiments in following studies.

- Please clarify why only 4 of the 6 coastal biofilm communities were chosen for metatranscriptome analysis, and how they were selected?

Reply: Thanks a lot for your careful checking. In the revised manuscript, we have added results of the other 2 metatranscriptomes, and the six metatranscriptomes have spanned all the seasons. The results indicated expression of *sox* genes in all the investigated metatranscriptomes.

- Please clarify why strain M382 was chosen for cultivation, gene knock-out, and proteomics?

Reply: Several reasons. First because this strain is likely to represent a new genus, and it has high abundance in surface-ocean biofilms and deep-sea biofilms. In addition, we had tried to knock out genes

in several other strains, and to the end only succeed in M382. Gene deletion in non-model strains is not an easy task. So, we only used M382 in the present study. However, we are working on other strains such as M597 and the results will be published in the near future.

- Was the transcriptomics experiment for strain M382 performed under aerobic or anerobic conditions, planktonically or as a biofilm? Would you detect difference in gene expression between planktonically and biofilm grown cells?

Reply: Good question! This transcriptomics experiment for strain M382 was performed in the biofilm state. We have described in the method part. In the revised manuscript, we added transcriptomics experiment for strain M382 living planktonically. As shown in **Figure for review 3** (please see this figure above), we found that when grown planktonically, thiosulfate could not induce the expression of *sox* genes. This result is interesting and suggests that the induction of *sox* genes by thiosulfate is biofilm specific. The transcriptomic data have been uploaded to the NCBI database, but not included in the revised manuscript, considering that this result has no impact on our central conclusion.

Ad (2) State of the art

- A quick browse of the recent literature shows that various pathways for thiosulfate oxidation are known in marine systems.

Reply: Good question! While *sox* genes are the most well-known pathways of thiosulfate oxidation, studies (Zhang et al., 2020) have suggested that thiosulfate dehydrogenase *tsdA* is also involved in the conversion of thiosulfate to zero-valent sulfur. We have added sentences to discuss the limitations of our work (please see the following). In addition, we searched (BLASTP, E-value < 1e-7 and similarity > 50%) the *tsdA* sequences in biofilm metagenomes and found no hit, suggesting that the abundance of this gene is actually very low in marine biofilms.

Limitations exist for the current study. For example, here we have focused on culturable *Roseobacter* strains in marine biofilms while metagenome-assembled genomes were not used. This is largely due to the high microbial diversity in marine biofilms, which hinders the recovery of high-quality genomes from biofilm metagenomes. Moreover, here we focused on the analyses of *sox* system in thiosulfate oxidation, which might be mediated by other genes (e.g., the thiosulfate dehydrogenase *tsdA* that works together with *soxB*).

Reference

Zhang, J., Liu, R., Xi, S., Cai, R., Zhang, X., & Sun, C. 2020. A novel bacterial thiosulfate oxidation pathway provides a new clue about the formation of zero-valent sulfur in deep sea. *The ISME journal*, 14, 2261-2274.

- Sulfitobacter, Rhodobacter and Erythrobacter are long known to perform aerobic and anaerobic sulfur oxidation, particularly in sediments, which are regarded as biofilms I believe.

Reply: Good question! The prevalent *Roseobacter* members in sediments are known to be *Sulfitobacter*, *Thalassobacter*, *Roseobacter*, and *Tateyamaria* (Lenk et al., 2012). However, no

Thalassobacter or *Tateyamaria* are detected in marine biofilms on stone surfaces, indicating that the taxonomic composition of *Roseobacter* members in biofilms (e.g., stone-surface biofilms) and those in sediments are significantly different. Moreover, in a recent study (under peer review in iMeta), we investigated biofilms on plastic and glass particles and found that biofilms on particles are totally different from those on large surfaces (e.g., plastic panels and stones). Considering that sediments are made of soil particles, biofilms on sediments are theoretically different from those on stone surfaces.

Reference

Lenk, S., Moraru, C., Hahnke, S., Arnds, J., Richter, M., Kube, M., ... & Mußmann, M. (2012). *Roseobacter* clade bacteria are abundant in coastal sediments and encode a novel combination of sulfur oxidation genes. *The ISME journal*, 6(12), 2178-2187.

- Is there evidence that thiosulfate is an abundant metabolite in marine waters/sediments?

Reply: Several studies have suggested thiosulfate to be an important and abundant metabolite in marine water and sediment, such as the work by Tuttle et al. (1977) and Jørgensen et al. (1990).

References

Tuttle, J. H., & Jannasch, H. W. (1977). Thiosulfate stimulation of microbial dark assimilation of carbon dioxide in shallow marine waters. *Microbial ecology*, 4, 9-25.

Jørgensen, B. B. (1990). A thiosulfate shunt in the sulfur cycle of marine sediments. *Science*, 249(4965), 152-154.

- Has it been shown before that addition of thiosulfate induces expression of the *sox* gene cluster?

Reply: We searched the literatures in Pubmed, and found a few studies that mentioned the induction of *sox* genes by thiosulfate, such as the work by Gwak et al. (2022), but none of these studies is related to *Roseobacter* strains.

Reference

Gwak, J. H., Awala, S. I., Nguyen, N. L., Yu, W. J., Yang, H. Y., von Bergen, M., ... & Rhee, S. K. (2022). Sulfur and methane oxidation by a single microorganism. *Proceedings of the National Academy of Sciences*, 119(32), e2114799119.

Ad (3) Proteomics conclusions

- Fueling biofilm development: There is no general biofilm concept, the mechanisms are different depending on ecosystem and strain, as I am sure you know. The reference that you cite for the energy demand for biofilm formation may not be relevant for marine biofilms. Likewise, many references can be found demonstrating that biofilms represent an attempt to survive harsh conditions, lack of nutrients etc. which means they reflect a condition where energy is scarce, and less is needed under biofilm conditions compared to planktonic life.

Reply: We agree with you. Because studies on marine biofilms are fewer than those on biofilms in other

environments (e.g., hospitals), energy for marine biofilm development is not well known. In the revised manuscript, by examining the PMF production in M382 biofilms with or without adding thiosulfate, we found that thiosulfate indeed contributes to energy production (**Extended Data Fig. 25** as shown above). Moreover, mutation of the *sox* genes reduces the cell density in M382 biofilms (**Extended Data Fig. 20** as shown above), also suggesting that thiosulfate oxidation contributes to energy production. In addition, the reason for biofilms representing an attempt to survive harsh conditions might be additional energy production in biofilms rather than no energy required in biofilms.

- Remodeling of the cell membrane: I think this is a mis-interpretation of the data. An enzyme can be localized either in the cytoplasm or in the cell membrane, but it is not recruited from the cytoplasm to the membrane if its substrate is provided.

Reply: We agree with you. Here our results suggested the change of protein compositions in cell membrane, but it could not be figured out whether these proteins can be relocalized. So we have revised such descriptions to avoid misleading.

- Thiosulfate treated cells: In what respect were the conditions different to those used for cultivation in ocean water with or without thiosulfate? In which respect were they different to the conditions used for the transcriptome analysis? Were the cells grown as biofilms or planktonically?

Reply: Good question! We used marine broth 2216 in proteomics and transcriptomics, because this medium is specific for the cultivation of marine bacteria. The salinity and inorganic elements in this medium is similar to those of the seawater, and it has been widely used in previous studies. In addition, considering that many *Roseobacter* strains are copiotrophic bacteria, it is probably that their natural features can be well reflected when cultured in marine broth 2216. Please refer to these publications in which marine broth 2216 is used to culture *Roseobacter* strains: Brinkhoff et al. 2004; Schaefer et al., 2002; Wang et al., 2022. The proteomics experiment for strain M382 was performed as a biofilm, under the same condition as the transcriptomics.

References

Brinkhoff, T., Bach, G., Heidorn, T., Liang, L., Schlingloff, A., & Simon, M. (2004). Antibiotic production by a *Roseobacter* clade-affiliated species from the German Wadden Sea and its antagonistic effects on indigenous isolates. *Applied and environmental microbiology*, 70(4), 2560-2565.

Schaefer, J. K., Goodwin, K. D., McDonald, I. R., Murrell, J. C., & Oremland, R. S. (2002). *Leisingera methylohalidivorans* gen. nov., sp. nov., a marine methylotroph that grows on methyl bromide. *International journal of systematic and evolutionary microbiology*, 52(3), 851-859.

Wang, M., Wang, H., Wang, P., Fu, H. H., Li, C. Y., Qin, Q. L., ... & Zhang, W. (2022). TCA cycle enhancement and uptake of monomeric substrates support growth of marine *Roseobacter* at low temperature. *Communications Biology*, 5(1), 705.

Minor comments:

- The taxonomy of roseobacters has been revised based on 106 completed genomes [1]. Please check if your phylogeny contains representatives from the main clusters described there.

1. Simon M, C Scheuner, JP Meier-Kolthoff, T Brinkhoff, I Wagner-Döbler, M Ulbrich, H-P Klenk, D Schomburg, J Petersen, M Göker. Phylogenomics of Rhodobacteraceae reveals evolutionary adaptation to marine and non-marine habitats. ISME J 2017; 11: 1483–1499.

Reply: Thanks for your recommended reference, and we have added it in the revised manuscript. In our tree, we included representative and well-studied members of the *Roseobacter* group, such as *Planktomarina temperate* RCA23 and *Ruegeria pomeroyi* DSS-3.

- “Roseobacter” is not a valid taxonomic term. Since roseobacters are not monophyletic, as previously assumed, the term “roseobacter group” should be used for marine Rhodobacteraceae.

Reply: Yes. In the revised manuscript, we used the term ‘*Roseobacter* group’ or ‘*Roseobacter* strains’. In a recent publication, roseobacters are recommended to be a new family designated ‘Roseobacteraceae’ and we have mentioned this in the introduction part.

Reference

Liang, K. Y., Orata, F. D., Boucher, Y. F., & Case, R. J. Roseobacters in a sea of poly- and paraphyly: whole genome-based taxonomy of the family Rhodobacteraceae and the proposal for the split of the “Roseobacter clade” into a novel family, Roseobacteraceae fam. nov. *Front. Microbiol.* **12**, 683109 (2021).

- Importance of plasmids: While you have detected plasmids with biofilm-relevant genes, you did not analyse if these genes are less abundant in genomes. So the conclusion that plasmids harbor the – biofilm related – diversity of roseobacters is not valid.

Reply: We agree with you. We have revised the statement in the discussion part. The copy number of biofilm-relevant genes in chromosomes shall be lower than those in plasmids. Unlike genes in chromosome, genes in plasmids can transfer among different strains. A previous study (Michael, et al. 2016) on one *Roseobacter* strain has suggested the importance of plasmid in biofilm formation. Combining genomic evidences in the present study, we conclude that biofilm-relevant genes in plasmids are diverse and important for *Roseobacter* strains in marine biofilms.

Reference

Michael, V., et al. Biofilm plasmids with a rhamnase operon are widely distributed determinants of the ‘swim-or-stick’ lifestyle in roseobacters. ISME J. 10(10), 2498-2513 (2016).

- Which carbon source was used for the cultivation in ocean water?

Reply: I think you mean the carbon source used in M382 transcriptomic and proteomic experiments, where we used marine broth 2216 that contains trypton and yeast extract as carbon source. As explained above, this medium is widely used in the study of marine microbes.

- Was M382 cultivated planktonically or as a biofilm? Was the growth effect of thiosulfate different under those two conditions?

Reply: M382 was cultivated as a biofilm, and the effect of thiosulfate on M382 was studied as a biofilm. We have added new transcriptomic data and analyses of M382 planktonically (please see **Figure for review 3** as mentioned above).

- How do you interpret the fact that the deletion mutants for *soxX* and *soxA* grew just as well, both aerobically and anaerobically, although they did not reduce thiosulfate?

Reply: Good question! We hypothesized that growth advantage of the wild-type strain might be related to the growth state (biofilm or free living). To test this hypothesis, we performed additional experiments to show that deletion of the *sox* genes can influence bacterial growth when cultured in biofilms rather than free-livingly (please see **Extended Data Fig. 20**). Consistently, thiosulfate oxidation contributes to energy production in biofilms rather than in free-living state (**Extended Data Fig. 25**).

Thanks again.
Weipeng

Reviewer #2 (Remarks to the Author):

I am pleased to report that the authors have addressed all of my questions and concerns. The manuscript has been carefully reviewed, and I believe it now meets the standards for publication. I appreciate the authors' thoroughness in responding to my comments, and I am confident that the revised manuscript will make a valuable contribution to the field. Thank you for the opportunity to review this work.

Reviewer #3 (Remarks to the Author):

Dear authors, thank you for revising the manuscript so carefully. My questions have all been answered satisfactorily. Good luck with your future work!